# EZH1 and EZH2 promote skeletal growth by repressing inhibitors of chondrocyte proliferation and hypertrophy

Julian C. Lui[1], Presley Garrison[1], Quang Nguyen[1], Michal Ad[1], Chithra Keembiyehetty[2], Weiping Chen[2], Youn Hee Jee[1], Ellie Landman[3], Ola Nilsson[3,4], Kevin M. Barnes[1] & Jeffrey Baron[1]

Histone methyltransferases EZH1 and EZH2 catalyse the trimethylation of histone H3 at lysine 27 (H3K27), which serves as an epigenetic signal for chromatin condensation and transcriptional repression. Genome-wide associated studies have implicated *EZH2* in the control of height and mutations in *EZH2* cause Weaver syndrome, which includes skeletal overgrowth. Here we show that the combined loss of Ezh1 and Ezh2 in chondrocytes severely impairs skeletal growth in mice. Both of the principal processes underlying growth plate chondrogenesis, chondrocyte proliferation and hypertrophy, are compromised. The decrease in chondrocyte proliferation is due in part to derepression of cyclin-dependent kinase inhibitors Ink4a/b, while ineffective chondrocyte hypertrophy is due to the suppression of IGF signalling by the increased expression of IGF-binding proteins. Collectively, our findings reveal a critical role for H3K27 methylation in the regulation of chondrocyte proliferation and hypertrophy in the growth plate, which are the central determinants of skeletal growth.

[1] Section on Growth and Development, Eunice Kennedy Shriver National Institute of Child Health and Human Development, National Institutes of Health, CRC, Room 1-3330, 10 Center Drive, MSC-1103, Bethesda, Maryland 20892, USA. [2] Genomic Core, National Institute of Diabetes and Digestive and Kidney Diseases, National Institutes of Health, Bldg8, Room 1A11, 8 Center Drive, Bethesda, Maryland 20892, USA. [3] Division of Pediatric Endocrinology. Q2:08, Department of Women's and Children's Health, Karolinska Institutet and University Hospital, 171 76 Stockholm, Sweden. [4] Department of Medical Sciences, Rm C1213, Örebro University and University Hospital, 701 85 Örebro, Sweden. Correspondence and requests for materials should be addressed to J.C.L. (email: luichunk@mail.nih.gov).

Longitudinal bone growth occurs at the growth plate. This cartilage structure consists of three histologically and functionally distinct layers, termed the resting zone (RZ), proliferative zone (PZ) and hypertrophic zone (HZ). Chondrocytes in the RZ serve as stem-cell-like precursors, which are capable of self-renewal and also give rise to clones of proliferative chondrocytes in the adjacent PZ. These clones are arranged in columns parallel to the long axis of the bone and undergo rapid proliferation. The chondrocytes subsequently stop dividing and enlarge to become the hypertrophic chondrocytes in the HZ. The HZ is invaded by blood vessels, osteoblasts and osteoclasts, which remodel the HZ cartilage into cancellous bone. This overall process, termed endochondral ossification, has been studied extensively because it drives bone elongation and therefore growth in overall body dimensions. In addition, the growth plate provides a powerful *in vivo* model for understanding tissue growth because, in the growth plate, unlike most other tissues, the progenitor cells, transit amplifying cells and terminally differentiated cells are spatially segregated into distinct zones, facilitating their individual study.

Findings suggest the importance of epigenetic mechanisms in regulating longitudinal bone growth. Mutations in multiple genes that encode DNA- and histone-modifying enzymes can cause skeletal overgrowth disorders. For example, mutations in *NSD1* or *SETD2*, both of which encode histone methyltransferases for H3K36, lead to Sotos or Sotos-like overgrowth syndrome[1–3]. Similarly, mutations in the DNA methyltransferase *DNMT3A* gene were reported to cause a distinct overgrowth syndrome with intellectual disability[4].

Another important chromatin modifier, the polycomb repressor complex 2 (PRC2), also regulates longitudinal bone growth. Comprised of four subunits, SUZ12, EED, RbAp48 and EZH1/EZH2, the PRC2 is responsible for catalysing the trimethylation of histone H3 at lysine 27 (H3K27me3)[5], which then serves as an epigenetic signal for chromatin condensation and transcriptional repression. In humans, heterozygous mutations in *EED* or *EZH2* cause Weaver[6,7] and Weaver-like syndrome[8], which are characterized by skeletal overgrowth, accelerated skeletal maturation and other skeletal abnormalities. In addition, the *EZH2* gene lies in a locus associated with adult human height variation[9,10], providing further evidence that *EZH2* has an important function in regulating skeletal growth.

In the current study, we utilize a mouse model with complete knockout of Ezh1 and cartilage-specific knockout of Ezh2 to explore the mechanisms by which PRC2 regulates skeletal growth. We show that PRC2 is important for both the proliferation and hypertrophy of growth plate chondrocytes. In the PZ, PRC2 suppresses the expression of Cdkn2a/b to allow normal cell cycle progression, while in the HZ, PRC2 suppresses Igfbp3/5 expression, thereby promoting IGF signalling and chondrocyte hypertrophy.

## Results

**Postnatal growth retardation in Ezh1/2 mice.** Consistent with prior studies, mice with complete knockout of Ezh1 were viable, fertile and showed no abnormalities in postnatal growth[11]. Similarly, Ezh1$^{-/-}$ Ezh2$^{fl/fl}$ mice (without cre), or cartilage-specific knockout of Ezh2 (Col2-cre Ezh2$^{fl/fl}$) in the presence of at least one copy of Ezh1 (Ezh1$^{+/-}$), were also viable, fertile and showed a postnatal growth pattern indistinguishable from wild-type (Col2-cre Ezh2$^{+/+}$) or heterozygous (Col2-cre Ezh2$^{+/fl}$) littermates (Supplementary Fig. 1).

Unlike the ubiquitous knockout of Ezh2, which is embryonic lethal[12], mice lacking both Ezh1 and Ezh2 in the cartilage (Col2-cre Ezh1$^{-/-}$ Ezh2$^{fl/fl}$, hereafter termed Ezh1/2 mice) were born at decreased frequency compared with the expected Mendelian ratio of 25% (Supplementary Table 1), but appears normal in size at birth. However, by 3 days after birth, the Ezh1/2 mice were noticeably smaller in size than wild-type mice, and the difference became more prominent with age (Fig. 1a,b). We speculate that this greater effect on skeletal growth after birth in some way reflects the major differences in the structure and function of the prenatal and postnatal growth plate cartilage[13]. Ezh1/2 mice showed increased mortality at ~2 weeks of age of apparent respiratory insufficiency. Whole-mount staining showed no obvious delay or advancement of skeletal maturation in the combined knockout, but a slightly hunched spine was observed at 3 days old near the distal end of the rib cage, which became more pronounced at 1–2 weeks of age (Fig. 1i–k). Decreased overall body growth (Fig. 1b) and longitudinal bone growth (Fig. 2i and see below) were observed as early as 3 days of age, when there were no differences in respiratory rate (Supplementary Fig. 2), suggesting that the decreased growth observed in the combined knockout is not solely a secondary effect of the respiratory insufficiency but instead that the respiratory insufficiency may be caused by insufficient growth of the rib cage and by the abnormal vertebral column.

Immunohistochemistry revealed a complete loss of Ezh2 expression (Fig. 1c,f) and dimethyl- and trimethyl-H3K27 (Fig. 1d,e,g,h) in the cartilage tissue of the combined knockout, but not in the surrounding perichondrium and adjacent bone tissues, suggesting that the Col2-promoter-driven Cre-recombination successfully abolished Ezh2 expression at the cartilage, and that Ezh1 and 2 redundantly function in establishing these epigenetic marks in chondrocytes. In addition, we also confirmed that Ezh1 protein is absent in these Ezh1/2 mice (Supplementary Fig. 3).

**Loss of Ezh1/2 leads to change in growth plate histology.** To understand better the role of Ezh1/2 in overall growth plate function, we performed quantitative histology in growth plates obtained from the proximal tibias of the Ezh1/2 and their wild-type and heterozygous littermates (Fig. 2a–f; Supplementary Fig. 4). The growth plate histology of heterozygotes was indistinguishable from wild-type littermates at the ages studied. In contrast, at 3 days of age, the overall tibial length, PZ height and the number of chondrocytes in the PZ were all significantly lower in the combined knockout (Fig. 2i).

In the combined knockout mice, the number of HZ chondrocytes per column was increased, but the height (dimension parallel to the long axis of the bone) of the terminal HZ chondrocyte was diminished (Fig. 2g–h,j; $P < 0.05$). The height of the terminal HZ chondrocyte plays an important role in determining the rate of longitudinal bone growth[14], and therefore the observed decrease in height of hypertrophic chondrocytes, which persisted in older mice (Fig. 2g,h,j), likely contributed to the impaired bone growth in Ezh1/2 mice.

To determine whether the smaller cells in the HZ of the Ezh1/2 mice might represent prehypertrophic chondrocytes, we used *in situ* hybridization to determine the location of Ihh (a marker for prehypertrophic chondrocytes) and ColX (a marker for hypertrophic chondrocytes) expression (Fig. 2k–r). We found that these smaller cells in the HZ of Ezh1/2 mice sequentially expressed Ihh then ColX in a pattern similar to that seen in wild-type mice, suggesting that, although most of the cells in the HZ are small in the Ezh1/2 mice, they are fully differentiated hypertrophic cells.

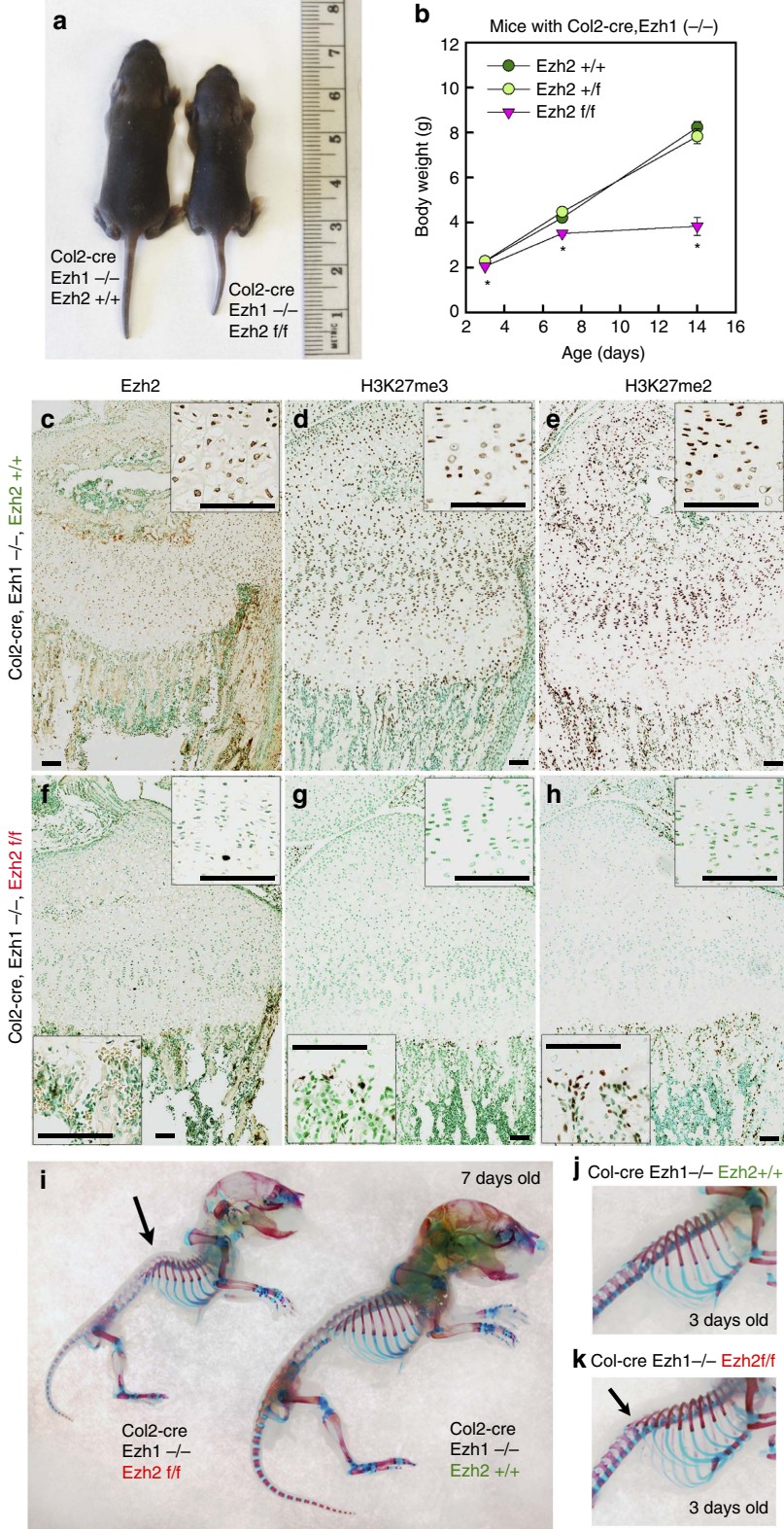

**Figure 1 | Ezh1/2 mice show growth retardation.** (**a**) One-week-old littermates of Ezh2 wild-type (Col2-cre, Ezh1 − / − , Ezh2 + / + ) and Ezh1/2 knockout (Col2-cre, Ezh1 − / − , Ezh2f/f) mice. (**b**) Growth curve of Ezh2 wild-type, cartilage-specific heterozygotes and cartilage-specific knockout mice, all with Ezh1 − / −  background. *P < 0.05 versus both wild type and heterozygotes within the same age group. (**c**–**h**) Proximal tibial growth plates of Ezh1 − / −  1-week-old mice with cartilage-specific knockout for Ezh2 (**d**,**f**,**h**) or wild type for Ezh2 (**c**,**e**,**g**) immunostained (brown colour) for Ezh2 (**c**,**d**), H3K27me3 (**e**,**f**) and H3K27me2 (**g**,**h**). Higher-magnification images of growth plate chondrocytes and spongiosa osteoblasts (for Ezh2f/f) were included to better illustrate the presence or absence of signal. (**i**–**k**) Whole-mount staining of Ezh1 − / −  mice with cartilage-specific knockout for Ezh2 or wild type for Ezh2 7 days (**i**) and 3 days (**j**,**k**) of age, stained for bone (Alizarin red) and cartilage (Alcian blue). Arrow indicates abnormal spinal curvature. Error bars, ± s.e.m. Scale bars, 100 μm.

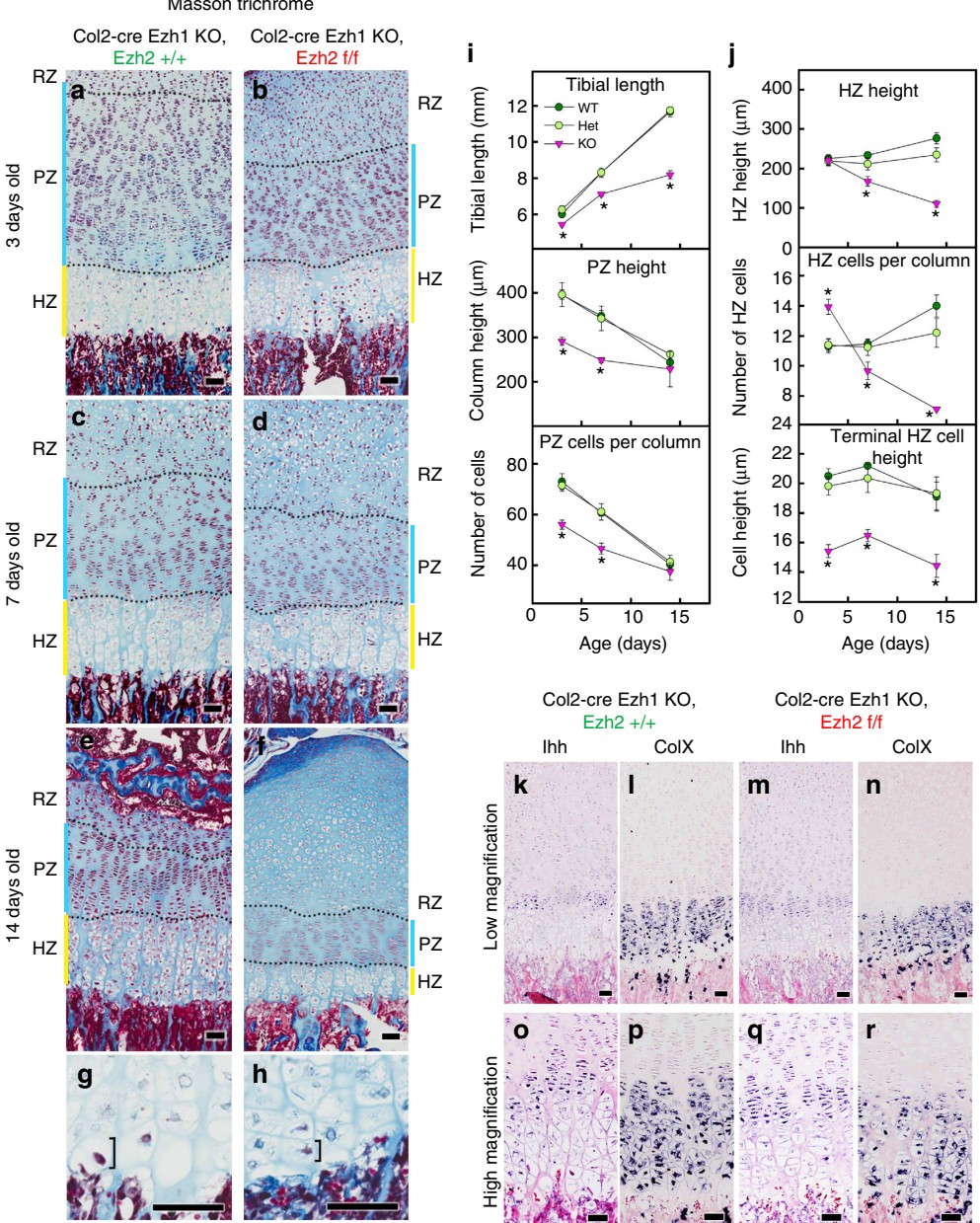

**Figure 2 | Quantitative histology of proximal tibial growth plate.** (**a–f**) Histological sections of proximal tibias of Ezh1 − / − mice with cartilage-specific knockout for Ezh2 (**b,d,f**) or wild type for Ezh2 (**a,c,e**) at 3 days (**a,b**), 1 week (**c,d**) and 2 weeks (**e,f**) of age. HZ, hypertrophic zone (yellow bars); PZ, proliferative zone (blue bars); RZ, resting zone. (**g,h**) Higher magnification of hypertrophic chondrocytes of Ezh1 − / − mice that are wild type (**g**) or with cartilage-specific knockout for Ezh2 (**h**) at 3 days old to illustrate the difference in terminal hypertrophic cell height (bracket). (**i,j**) Quantitative histological measurements of tibial length, PZ height and number of cells per proliferative column (**i**), and HZ height, number of cells per hypertrophic column and terminal hypertrophic cell height (**j**) in Ezh1 − / − mice that are wild type (WT, $N = 6$), heterozygous (Het, $N = 6$) or with cartilage-specific knockout (KO, $N = 8$) for Ezh2. *$P < 0.05$ versus both WT and Het within the same age group. (**k–r**) In situ hybridization of Ihh (**k,m,o,q**) and ColX (**l,n,p,r**) of proximal tibias of Ezh1 − / − mice with cartilage-specific knockout for Ezh2 (**m,n,q,r**) or wild type for Ezh2 (**k,l,o,p**) at 1 week of age. Error bars, ± s.e.m. Scale bars, 50 μm.

**Decreased chondrocyte proliferation in Ezh1/2 mice.** We next sought to elucidate the mechanisms that led to the observed changes in the PZ. The Ezh1/2 mice showed decreased BrdU labelling in the proliferative columns at all ages compared with wild-type and heterozygous littermates (Fig. 3a–d), indicating impaired cell proliferation. A similar decrease in proliferation was observed when primary chondrocytes were isolated from these Ezh1/2 mice and cultured in monolayer (Fig. 3e), suggesting that the effect of Ezh1/2 deletion on proliferation is likely due to a cell-autonomous mechanism.

**Cdkn2a/b upregulation contributes to proliferation defects.** The *INK4B-ARF-INK4A* genomic locus, which contains cyclin-dependent kinase (CDK) inhibitor p16[INK4a] and p15[INK4b] (encoded by Cdkn2a and 2b), inhibits cell proliferation and is a recognized molecular target of the PRC2 (ref. 15). We therefore hypothesized that the loss of Ezh1 and 2 allows derepression of Cdkn2a and 2b in the PZ, thus contributing to the diminished proliferation rate. Because each zone of the growth plate has a highly distinct profile of gene expression[16], we used laser capture microdissection (LCM) to isolate the three different zones

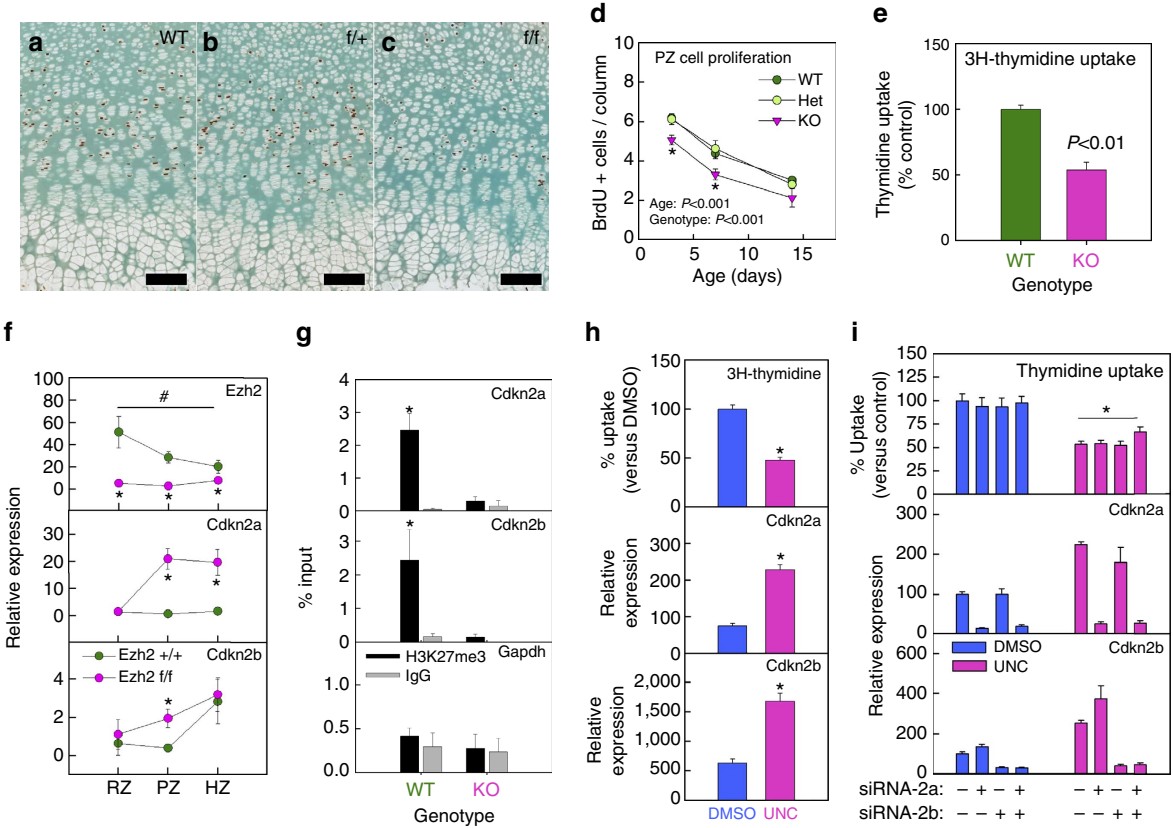

**Figure 3 | Cdkn2a/b upregulation contributed to defects in chondrocyte proliferation.** (**a**–**c**) BrdU staining of 1-week-old proximal tibial growth plate from Ezh1 − / − mice that are wild type (WT) (**a**), heterozygous (**b**) or homozygous (**c**) for cartilage-specific Ezh2 knockout. (**d**) Number of BrdU-positive cells per proliferative column in proximal tibial growth plates of 3-day-, 1-week- and 2-week-old Ezh1 − / − mice that are WT (N = 6), heterozygous (HET, N = 6) or homozygous (KO, N = 8) for cartilage-specific Ezh2 knockout. *P < 0.05 versus both WT and Het within the same age group. (**e**) Tritiated thymidine uptake by monolayer primary chondrocytes, isolated from 1-week-old Ezh1 − / − mice that are WT (N = 6) or homozygous (KO, N = 8) for cartilage-specific Ezh2 knockout (E). (**f**) Relative expression of Ezh2, Cdkn2a and Cdkn2b in different zones (RZ, PZ and HZ) isolated from proximal tibial growth plates of 3-day-old Ezh1 − / − that are WT (Ezh2 + / + ) or homozygous (Ezh2 fl/fl) for cartilage-specific Ezh2 knockout. Tissue was isolated by LCM and messenger RNA measured by quantitative real-time PCR; #P < 0.05 across zones in Ezh2 WT; *P < 0.05 between Ezh2 WT and Ezh1/2 knockout within the same zone. (**g**) Chromatin immunoprecipitation (ChIP) with H3K27me3 antibody (or IgG), followed by real-time PCR to compare levels of histone modification near transcription start site of Cdkn2a, Cdkn2b and Gapdh, between chondrocytes isolated from 1-week-old Ezh1 − / − mice that are WT or homozygous (KO) for cartilage-specific Ezh2 knockout. *P < 0.05, WT H3K27me3 versus KO H3K27me3. (**h**) Tritiated thymidine uptake and Cdkn2a and Cdkn2b expression (by real-time PCR), in monolayer primary chondrocytes isolated from 1-week-old WT mice, treated with Ezh1/2 inhibitors (UNC) or DMSO; *P < 0.05, N = 6. (**i**) Similar experiment as in **h**, except that chondrocytes were treated with siRNA against Cdkn2a and/or Cdkn2b (or scrambled siRNA) before Ezh1/2 inhibition; *P < 0.05, N = 6. Error bars, ± s.e.m. Scale bars, 100 μm.

(RZ, PZ and HZ) from 3-day-old tibial growth plate, extracted RNA and performed quantitative real-time PCR (qPCR). Growth plate chondrocytes from Ezh1/2 mice showed much lower expression of Ezh2 in all three zones compared with wild-type littermates, further confirming the efficient cre-mediated ablation of the Ezh2 gene (Fig. 3f). Interestingly, Ezh2 showed a decline in gene expression across the wild-type growth plate from RZ to HZ (Fig. 3f). Expression of Cdkn2a was low throughout the wild-type growth plate, but was ~20-fold higher in the PZ and HZ of the Ezh1/2 mice (Fig. 3f). Cdkn2b expression was also increased in Ezh1/2 mice compared with wild-type mice, but primarily only in the PZ (Fig. 3f). We then used immunohistochemistry to confirm that the protein levels of Cdkn2a and 2b were similarly increased in the Ezh1/2 mice (Supplementary Fig. 5). These data support our hypothesis that Cdkn2a and 2b were derepressed in the growth plates of Ezh1/2 mice. We next isolated chondrocytes from Ezh1/2 mice and their wild-type littermates to compare their H3K27me3 levels at the *INK4A/B* locus. Chromatin immunoprecipitation (ChIP)–qPCR showed that H3K27me3 was significantly enriched at the promoter regions of both

Cdkn2a and 2b in primary chondrocytes isolated from wild-type littermates, but not from the Ezh1/2 mice (Fig. 3g). Our data therefore suggest that, in wild-type chondrocytes, this genomic locus is epigenetically suppressed, while in Ezh1/2 knockout chondrocytes, the loss of Ezh1/2 expression led to decreased H3K27me3 and upregulation of Cdkn2a/b. To further establish the causal relationship between Ezh1/2 deletion, Cdkn2a/b upregulation and decreased proliferation in chondrocytes, we next treated primary chondrocytes isolated from wild-type C57BL/6 mice with inhibitors of Ezh1/2 and measured cell proliferation and Cdkn2a/b expression. Consistent with our hypothesis, Ezh1/2 inhibitor UNC1999 increased expression of both Cdkn2a and 2b, and suppressed chondrocyte proliferation (Fig. 3h). To confirm that the decrease in chondrocyte proliferation is driven by elevated Cdkn2a/b levels, we next used short interfering RNA (siRNA) to suppress Cdkn2a/b expression before UNC1999 treatment. Knockdown of Cdkn2a/b expression was confirmed by qPCR (Fig. 3i) and western blot (Supplementary Fig. 6). In the absence of UNC1999, suppression of Cdkn2a/b expression by siRNA alone did not

increase chondrocyte proliferation (Fig. 3i), again suggesting that Cdkn2a/b was normally silenced in chondrocytes. In contrast, treatment of UNC1999 led to the suppression of chondrocyte proliferation, which was then partially reversed by siRNA against Cdkn2a/b (Fig. 3i). Our data therefore suggest that the decrease in chondrocyte proliferation induced by loss or inhibition of Ezh1/2 is partially (but not completely) attributable to derepression of Cdkn2a/b expression, but that other molecular pathways may also make important contributions to the impaired proliferation.

**Increased Igfbp3 and 5 expression in Ezh1/2 mice.** In addition to decreased proliferation in the PZ, another remarkable finding in the Ezh1/2 mice was the decreased terminal hypertrophic cell size (Fig. 2g,h). Because insulin-like growth factor (Igf) signalling is required for normal chondrocyte hypertrophy[17,18], we hypothesized that the effect of Ezh1/2 loss on hypertrophic cell size is mediated by diminished Igf1 signalling.

To test our hypothesis, we first assessed the expression of genes in the Igf signalling pathway in the HZ of Ezh1/2 mice and wild-type littermates. We found that Igf2, rather than Igf1, is the predominant ligand in the HZ[19], but the expression of neither ligand was significantly different between wild-type and Ezh1/2 mice (Supplementary Fig. 7). Similarly, the two specific receptors for Igfs, Igf1r and Igf2r, were not differentially expressed between wild-type and Ezh1/2 mice (Supplementary Fig. 7). However, we found that two members of the Igf-binding protein (Igfbp) family, Igfbp3 and 5, were significantly upregulated specifically in the HZ of the Ezh1/2 mice compared with wild-type littermates (Fig. 4a), which were confirmed at the protein level by immunohistochemistry (Supplementary Fig. 5). H3K27me3 was significantly enriched at the promoter region of both Igfbp3 and 5 in chondrocytes isolated from wild-type littermates but not in Ezh1/2 mice (Fig. 4b), suggesting that the effect on expression is mediated by H3K27 methylation. In contrast, no H3K27me3 enrichment was found at the promoter region of Ihh or ColX (Supplementary Fig. 8).

**Ezh inhibition upregulates Igfbps and decreases hypertrophy.** The observed increase in the expression of Igfbps in chondrocytes could reduce the bioavailability of Igfs and therefore might contribute to the diminished hypertrophy. We tested this hypothesis further using monolayer cultures of primary wild-type chondrocytes. When these cells were treated with Ezh1/2 inhibitor UNC1999, Igfbp3, although not Igfbp5, increased (Fig. 4c). We next used a chondrocyte pellet culture model to study chondrocyte hypertrophy. Primary chondrocytes isolated from wild-type C57BL/6 mice were pelleted by centrifugation (Fig. 4d) and maintained in culture for up to 4 weeks. Chondrocyte pellets normally undergo hypertrophy with time in culture, leading to the increase in cell size (Fig. 4d) and expression of differentiation markers including Col10a1 and Ihh (Fig. 4g). Treatment of the pellets with Ezh1/2 inhibitor UNC1999-treated reduced levels of H3K27me2 and me3 (Fig. 4e,f), remarkably reduced cell hypertrophy (Fig. 4d), and decreased the expression of Col10a1 and Ihh (Fig. 4g). Consistent with our hypothesis, Igfbp5 expression was increased in UNC1999-treated pellets (Fig. 4g). In contrast, Igfbp3 was not well expressed in either treated or untreated pellets in this experimental model.

**Igfbp upregulation contributes to decreased hypertrophy.** To further establish the causal relationship between Ezh1/2 deletion, Igfbp3/5 upregulation and decreased chondrocyte hypertrophy, we next utilized a well-established metatarsal culture system to test the effect of Ezh1/2 inhibition on Igf signalling[20]. Fetal metatarsal bones were isolated from wild-type C57BL/6 mice at

embryonic day 18 and cultured for 3 days (Fig. 4h; Supplementary Fig. 9). Treatment with UNC1999 decreased H3K27me2 and me3 (Fig. 4j; Supplementary Fig. 10), and increased the expression of Igfbp3 and 5 in the metatarsal growth plate (Fig. 4i). This upregulation of both Igfbp3 and 5 is similar to the upregulation of both genes observed in the Ezh1/2 mice, and thus the cultured metatarsal bones may serve as a more physiological model system than either the monolayer culture, in which only Igfbp3 was upregulated, or the pellet culture, in which only Igfbp5 was upregulated upon UNC treatment. Addition of UNC1999 resulted in a reduction of hypertrophic cell size (Fig. 4l,m), similar to that observed in the HZ of Ezh1/2 mice (Fig. 2h), and the UNC1999-treated chondrocyte pellets (Fig. 4d). In contrast, treatment with Igf1 for 3 days led to significantly increased bone growth (Fig. 4k) and chondrocyte hypertrophy in the fetal metatarsal (Fig. 4l,m). If the effect of Ezh2 on hypertrophy is mediated by increased binding of Igfs by Igfbps, then adding a high concentration of Igf1 should exceed the amount that can be bound by endogenous Igfbps and thus override the inhibitory effect on hypertrophy. Consistent with this prediction, Igf1 treatment overrode the effect of UNC1999 in that hypertrophic cell size was similarly increased in metatarsal bones treated with Igf1 alone and treated with both Igf1 and UNC1999 (Fig. 4l,m). In contrast, Igf1 treatment did not significantly affect chondrocyte proliferation (Supplementary Fig. 11). As an additional test of our hypothesis, we cultured the fetal metatarsal bones with or without UNC1999, and treated with exogenous Igfbp3 or Igfbp5 in high concentration for 3 days (Fig. 4n; Supplementary Fig. 9). In the absence of exogenous Igfbp3 or 5, addition of UNC1999 again resulted in the reduction of hypertrophic cell size (Fig. 4o,p). Importantly, we found that the treatment of exogenous Igfbp3 or 5 alone led to a similar reduction of hypertrophic cell size, with no additional effect when metatarsals were treated with both UNC1999 and Igfbp3 or 5 (Fig. 4o,p), supporting our hypothesis that the loss of Ezh1/2 decreased chondrocyte hypertrophy by increasing levels of Igfbps. Taken together, our findings strongly support the hypothesis that Ezh1/2 positively regulates Igf signalling in the HZ to promote chondrocyte hypertrophy by negatively modulating local levels of Igfbp3 and 5.

**Transcriptome analysis of the growth plate in Ezh1/2 mice.** To explore other potential pathways by which the loss of Ezh1 and 2 may affect growth plate chondrogenesis, we used microarray to analyse differences in gene expression in the growth plate between Ezh1/2 and wild-type mice. We used LCM to isolate growth plate tissue from either the PZ or HZ, isolated RNA, and performed microarray analysis and pathway analysis. We found that 49 genes were downregulated ($P < 0.001$, fold change $> 1.3$) in PZ of Ezh1/2 mice, while 231 genes were upregulated compared with wild type. Similarly in HZ, 97 genes were downregulated ($P < 0.001$, fold change $> 1.3$) and 146 genes were upregulated in the Ezh1/2 mice, compared with the wild-type mice (Fig. 5a,c; Supplementary Data 1 and 2). To confirm the validity of the microarray analysis, we performed real-time PCR on a subset of genes that showed differential regulation (Fig. 5e). The observation that more genes were upregulated in Ezh1/2 mice (as opposed to downregulated), both in PZ and HZ, is consistent with the role of Ezh1 and 2 in catalysing H3K27 trimethylation, which serves as a repressive mark of gene expression. Interestingly, the number of genes that were upregulated or downregulated concordantly in both the PZ and HZ was modest (Fig. 5b). Thus, the targets of H3K27 repression appear to depend on the differentiation state of the chondrocytes. The microarrays results also confirmed the upregulation of

Cdkn2a/b in the PZ and the upregulation of Igfbp3/5 in the HZ (Supplementary Data 1 and 2). Next, we used Ingenuity Pathway Analysis to explore additional potential signalling pathways (Fig. 5d). As expected, this analysis implicated cell cycle G1/S regulation in the PZ (which includes the upregulation of Cdkn2a/b) and implicated IGF signalling in the HZ (which includes the upregulation of Igfbp3/5). In addition, the analysis implicated Wnt/β-catenin and glucocorticoid receptor signalling in the PZ and thyroid hormone receptor and retinoid signalling in the HZ, all of which are thought to play important roles in the regulation of growth plate chondrocytes (Fig. 5d; Supplementary Data 3 and 4).

## Discussion

Recent human genetic findings indicate that epigenetic modifications, including histone methylation, are important

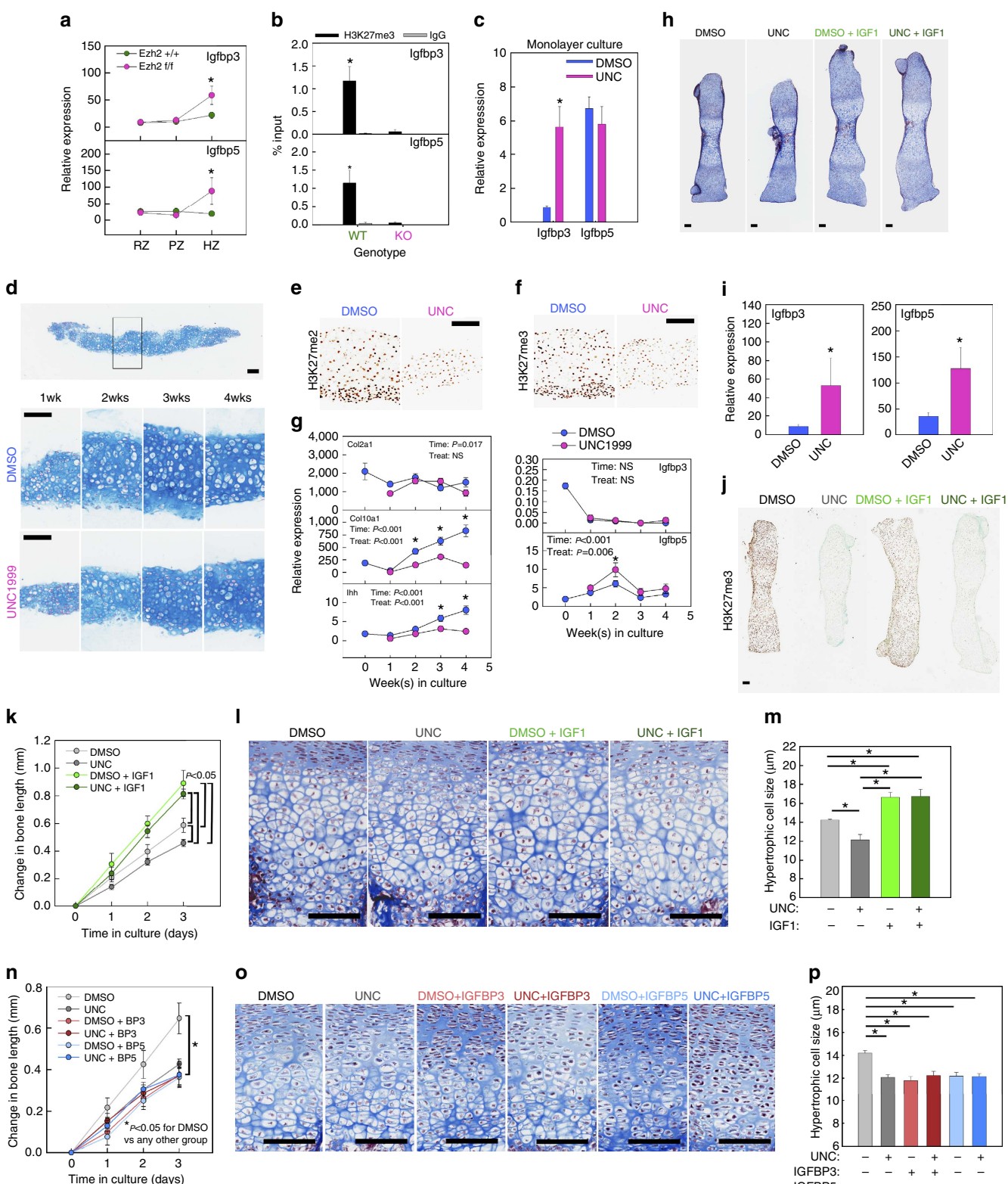

regulators of skeletal growth. However, the mechanisms remain largely unknown. In the current study, we showed that the combined loss of histone methyltransferases Ezh1 and Ezh2 in cartilage resulted in a loss of PRC2 methyltransferase activity in the growth plate and a severe impairment in longitudinal bone growth. The deficient skeletal growth was attributable to a reduction in both of its principal determinants[14], growth plate chondrocyte proliferation and hypertrophy. We then went on to elucidate the molecular mechanisms responsible for these cellular effects.

In the PZ, we found that Ezh1/2 normally promotes proliferation in part by repressing CDK inhibitor Cdkn2a/p16[INK4a] and Cdkn2b/p15[INK4b] (encoded by *Cdkn2a* and *Cdkn2b*; Fig. 6). The *INK4B-ARF-INK4A* locus has previously been shown to be a key target of Ezh2 in other cells, including epidermal stem cells[11], pancreatic beta cells[21], muscle satellite cells[22], haematopoietic stem cells[23] and numerous cancer cells[24,25]. The PRC2 binds to this genomic locus to initiate H3K27 trimethylation and transcriptional silencing of these CDK inhibitors, which in turn helps promote cell cycle progression and cell proliferation[15].

In the HZ, we found that Ezh1/2 normally promotes cell hypertrophy by repressing the expression of Igfbp3 and 5 (Fig. 6). Downregulation of Igfbp3 and 5 by Ezh2 has also been observed in other cell types[26–28]. These extracellular proteins bind IGF-I and -II, decreasing their ability to interact with the insulin-like growth factor I receptor, which is required for normal chondrocyte hypertrophy. Thus, by downregulating Igfbp3 and 5 expressions, Ezh1/2 allows active IGF signalling and consequently chondrocyte hypertrophy.

Our findings also reveal that Ezh1 and 2 possess compensatory methyltransferase activity in the growth plate. The presence of any one wild-type allele of either Ezh1 or 2 (such as Col2-cre Ezh1$^{-/-}$ Ezh2$^{+/fl}$ and Col2-cre Ezh1$^{+/-}$ Ezh2$^{fl/fl}$) is sufficient to maintain H3K27 methylation and normal longitudinal bone growth. This functional redundancy has also been observed in hepatocytes but is not universal; loss of Ezh2, but not Ezh1, causes embryonic lethality[29]. The ability of each methyltransferase to compensate for the other in a specific tissue likely depends on expression levels of the two genes in that tissue[30] and on functional differences between the two enzymes[30].

Our observations also provide interesting insights into the pathophysiology of Weaver syndrome, which can be caused by heterozygous missense mutation in *EZH2* (refs 6,7). Our data suggest that the mutations causing Weaver syndrome are not simple loss-of-function mutations because, in mice, heterozygous—or even homozygous—loss-of-function mutations in *Ezh2* do not have a skeletal phenotype unless *Ezh1* is also ablated. Furthermore, individuals with Weaver syndrome show tall stature due to increased skeletal growth, whereas, in mice, deletion of Ezh1/2 leads to diminished skeletal growth. Although these apparent discrepancies could be due to species differences, our data suggest that Weaver mutations may cause a gain-of-function of the PRC2. This possible explanation is not supported by a recent study[31] showing evidence that Weaver mutations impaired methyltransferase activity *in vitro*. However, EZH2 catalyses three sequential methyltransferase reactions at H3K27 to produce H3K27me3, and mutations can cause complex effects on substrate preference, which, in combination with wild-type enzyme from the other allele, may produce hyper-trimethylation[32]. Therefore, full assessment of the functional consequences of *EZH2* mutations is complex. Thus, the molecular pathophysiology of *EZH2* mutations in Weaver syndrome requires further investigation.

Several prior studies have investigated the role of Ezh2 and H3K27 methylation in growth plate chondrogenesis. Dudakovic et al.[33] studied a mouse with Ezh2 ablated in uncommitted mesenchymal cells. Thus, their model differed from ours in that the loss of Ezh2 occurred earlier in cell differentiation and Ezh1 was left intact. They observed shortened long bones and vertebrae, reduced HZ height and overall growth plate height. Zhang et al.[34] studied Jmjd3, which encodes an H3K27me3 demethylase, and found that ablation of the gene in mice decreases proliferation and delays hypertrophy of chondrocytes, indicating that increased H3K27 methylation, like the decreased methylation in the current study, can severely impair endochondral bone formation. Very recently, Mirzamohammadi et al.[35] described a mouse with cartilage-specific loss of Eed, which like Ezh2, is a component of the PRC2 that catalyses the trimethylation of H3K27. These mice showed the effects similar to Ezh1/2 mice with kyphosis, decreased chondrocyte proliferation, accelerated hypertrophic differentiation and cell death with reduced Hif1a expression. Their study focused on the roles of excessive Wnt and TGF-β signalling.

In conclusion, our findings indicate that histone methyltransferases Ezh1 and 2 are jointly required for normal skeletal growth. Ablation of both genes in growth plate inhibits both chondrocyte proliferation and hypertrophy. The effect of the PRC2 on proliferation appears to be mediated in part by silencing of Cdkn2a and 2b expression, whereas the effect on cell hypertrophy is mediated, at least in part, by silencing of Igfbp3 and 5 expression, thus augmenting Igf availability.

## Methods

**Animals.** All animal procedures were approved by the National Institute of Child Health and Human Development Animal Care and Use Committee. Col2-cre mice

---

**Figure 4 | Igfbp3/5 upregulation contributed to defects in chondrocyte hypertrophy.** (**a**) Relative expression of Igfbp3 and 5 in different zones (RZ, PZ and HZ) isolated from proximal tibial growth plates of 3-day-old Ezh1/2 or wild-type mice. Tissue was isolated by laser capture microdissection, and messenger RNA levels were measured by real-time PCR; *$P < 0.05$ between Ezh1/2 or wild-type mice within the same zone ($N = 6$). (**b**) ChIP with H3K27me3 antibody (or IgG), followed by real-time PCR to compare levels of H3K27me3 near transcription start site of Igfbp3 and Igfbp5, in chondrocytes isolated from 1-week-old Ezh1/2 or wild-type mice. *$P < 0.05$ between Ezh1/2 or wild-type mice for H3K27me3 ($N = 6$). (**c**) Relative expression of Igfbp3 and Igfbp5 in monolayer primary chondrocytes isolated from 1-week-old wild-type mice treated with an Ezh1/2 inhibitor (UNC) or vehicle (DMSO). *$P < 0.05$ ($N = 6$). (**d**) Top panel: Alcian blue-stained histological section of a chondrocyte pellet cultured for 1 week. Middle and bottom panels: higher magnification of DMSO-treated pellets (middle panel) or UNC-treated pellets (bottom panel) at different time points. (**e,f**) Histological sections of pellet treated with DMSO or UNC for 1 week, immunostained (brown colour) for H3K27me2 (**e**) or H3K27me3 (**f**). (**g**) Relative expression of Col2a1, Col10a1, Ihh, Igfbp3 and Igfbp5 in chondrocyte pellets treated with DMSO or UNC at different time points. *P* values, two-way analysis of variance for the effect of time in culture and UNC treatment. *$P < 0.05$ between DMSO and UNC treated at a particular time point ($N = 6$). (**h**) Histological sections of cultured fetal mouse metatarsal bones treated with DMSO or UNC, with or without IGF-I. (**i**) Relative expression of Igfbp3 and Igfbp5 in metatarsal whole growth plate from **h**. *$P < 0.05$. (**j**) Immunohistochemistry (brown colour) for H3K27me3 of metatarsal bones from **h**. (**k–m**) Change in length (**k**) histological sections (**l**), and quantitative measurements of hypertrophic cell size (**m**) in fetal metatarsal bones treated with DMSO or UNC, with or without Igf1. Statistical comparison was performed on bone length between different treatment groups at the end of treatment (**m**). (**n–p**) Similar to **k–m**, except for the treatment with Igfbp3/5 instead of IGF-I. *$P < 0.05$, $N = 6$ (**m,p**). *$P < 0.05$, DMSO versus all other groups (n). Scale bars, 100 μm.

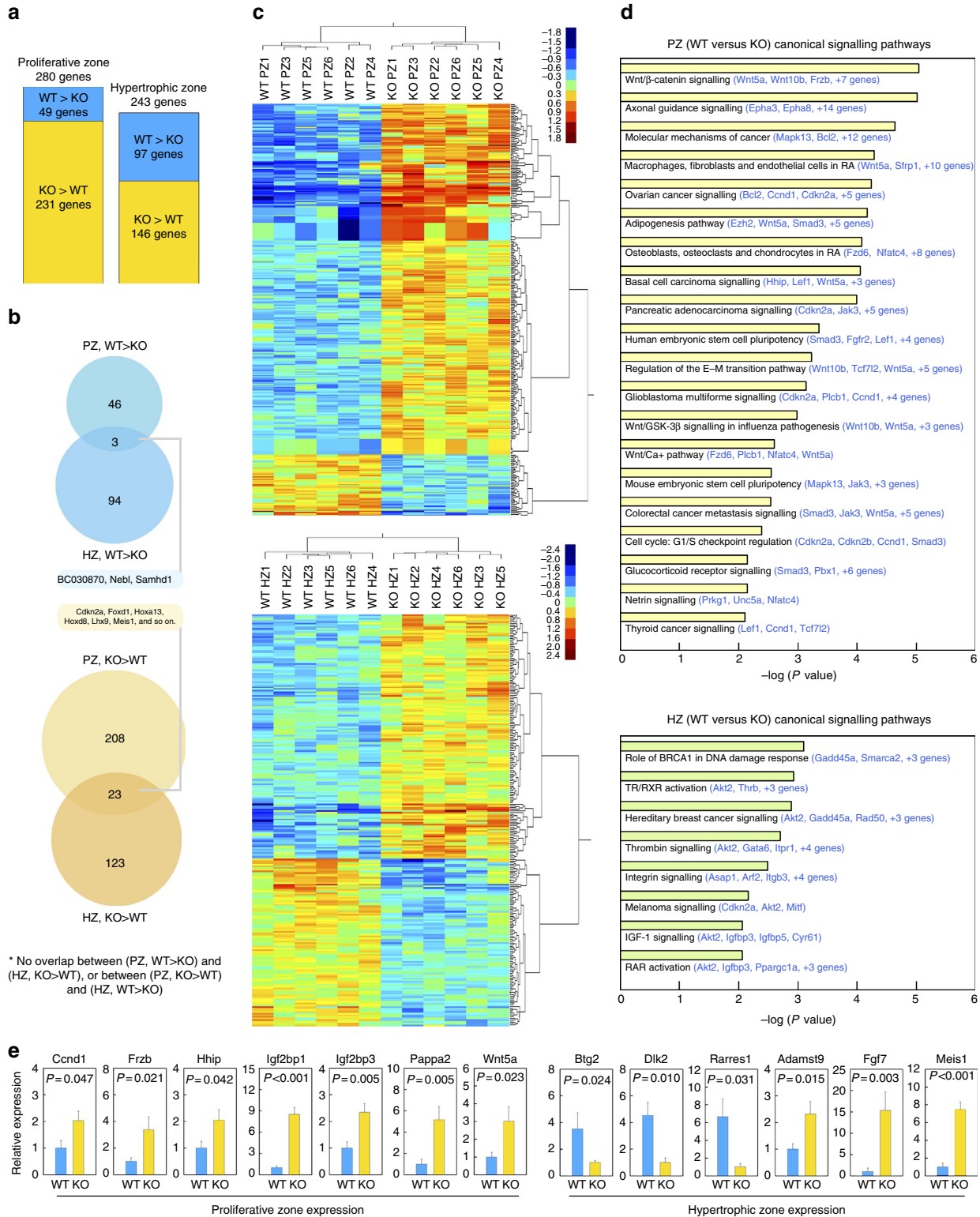

**Figure 5 | Microarray of gene expression in growth plates of Ezh1/2 mice.** Expression microarray was performed in RNA isolated from tibial growth plate proliferative or hypertrophic zone of 3-day-old Ezh1/2 or wild-type mice using laser capture microdissection. In both zones, using a cutoff of $P < 0.001$ and fold change $> 1.3$, more genes were upregulated than downregulated in Ezh1/2 mice compared with wild type (**a**). Overlap of genes commonly upregulated (or commonly downregulated) between two zones in the Ezh1/2 mice were modest (**b**). Unsupervised two-way hierarchical clustering of the 12 samples from Ezh1/2 mice and wild-type mice (six samples each), using the 280 significant genes in the PZ (top panel) and 243 significant genes in the HZ (bottom panel). Relative expression was depicted by colour maps, with blue representing low expression, and red representing high expression. Scale bar, normalized $\log_2$-transformed expression (**c**). Canonical signalling pathway analysis by Ingenuity Pathway Analysis on the 280 PZ genes (top panel) and 243 HZ genes (bottom panel) (**d**). A subset of messenger RNAs that were found to be differentially expressed in Ezh1/2 versus wild-type mice were analysed by real-time PCR for confirmation (**e**). Error bars, $\pm$ s.e.m.

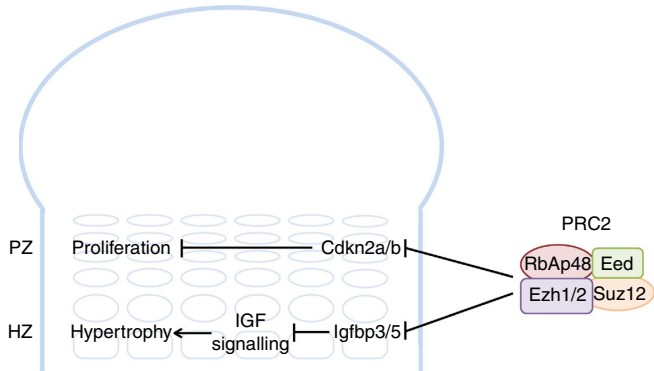

**Figure 6 | PRC2 promotes chondrocyte proliferation and hypertrophy.**
Our findings support this working model to explain how PRC2 promotes proliferation and hypertrophy in the growth plate. In the proliferative zone (PZ), PRC2 suppresses expression of Cdk inhibitors Cdkn2a and 2b, to allow normal cell cycle progression and proliferation. In the hypertrophic zone (HZ), PRC2 positively modulates IGF signalling by suppressing the expression of Igfbp3 and 5, therefore promoting chondrocyte hypertrophy.

were kindly provided by Dr Susan Mackem from the National Cancer Institute, Frederick, Maryland. Ezh1 knockout and Ezh2 flox mice[29] were kindly provided by Dr Lothar Hennighausen from the National Institute of Diabetes and Digestive and Kidney Diseases, Bethesda, Maryland. The three lines of mice were crossed to obtain Col2-cre Ezh1 $^{-/-}$ Ezh2 $^{+/fl}$. These mice, on a mixed background, were then mated to generate mice wild type, heterozygous and homozygous for the Ezh2 flox allele. In addition, C57BL/6 mice were obtained from Charles River Laboratory. A combination of male and female mice was used in our experiments.

**Whole-mount preparation.** Whole-mount staining of bone and cartilage with Alizarin red and Alcian blue was performed as previously described[36]. Briefly, after removing skins, internal organs and adipose tissue, the mouse body was fixed in 95% ethanol (overnight), followed by acetone (overnight) and stained with Alcian blue solution (0.03% (w/v)). Specimens were then destained by 70% ethanol and 95% ethanol (overnight), and pre-cleared with 1% KOH solution. Specimens were then stained with Alizarin red (0.005% (w/v)) for 2 h before destaining in 1% KOH solution. Specimens were then stored in 100% glycerol.

**Growth plate dissection.** After mice were killed, distal femur and proximal tibial epiphyses were excised. For LCM, cartilage was embedded in optimum cutting temperature compound (Electron Microscopy Sciences, Hatfield, PA), frozen on dry ice and stored at –80 °C. LCM of growth plate cartilage was performed as previously described[37]. For histology and immunostaining, cartilage was fixed overnight in 2% (w/v) paraformaldehyde and 0.2% (v/v) glutaraldehyde at 4 °C and decalcified in 10% (w/v) EDTA and 0.5% (w/v) paraformaldehyde, pH 7.4. Samples were then embedded in paraffin for sectioning. Histological sections, included Masson's trichrome- and Alcian blue-stained sections, were prepared by Histoserv (Germantown, MD).

**Growth plate quantitative histology.** Histological evaluations were performed on Masson's trichrome-stained proximal tibia sections, visualized using a ScanScope CS digital scanner (Aperio Technologies, Inc) under bright-field microscopy. All histological measurements were performed in the central two-third of the proximal tibia growth plate sections. Heights were measured parallel to the chondrocyte columns. Column density was calculated as the number of columns per 500 μm growth plate width. Hypertrophic chondrocytes were operationally defined by a height ≥10 μm (ref. 38). The terminal hypertrophic chondrocyte was defined as the cell in the last lacuna that was not invaded by metaphyseal blood vessels. For each growth plate section, we performed at least three measurements of PZ height, HZ height and terminal hypertrophic cell height. For terminal hypertrophic cell height, the height of the lacunae, which reflects the actual hypertrophic chondrocytes height before cells condense during tissue fixation and processing, were measured. We also counted the number of proliferative and hypertrophic cells in at least three intact columns from each growth plate section. For each animal, averages were taken on a total of at least 24 different measurements from eight growth plate sections.

**BrdU staining.** The proliferation rate was determined by 5-bromo-2-deoxyuridine (BrdU) staining. BrdU was injected (0.1 mg g $^{-1}$ body mass intraperitoneal, Sigma-Aldrich, St Louis, MO) 2 h before mice were killed, and the growth plates were dissected, fixed and decalcified. Samples were embedded in paraffin, and 10 μm

sections were mounted on Superfrost Plus slides. BrdU labelling was detected by immunohistochemistry using the BrdU *In Situ* Detection kit (BD Biosciences, San Jose, CA) and counter-stained with methyl green. BrdU-positive cells were counted in the centre two-third of the growth plate PZ, and divided by the number of proliferative columns in the counted area.

**Immunostaining.** Growth plate sections were baked at 65 °C for 1 h, deparaffinized in xylene, rehydrated through ethanol series (100, 100, 95 and 70%) and rinsed with PBS. For all antibodies except Igfbp5, antigen retrieval was performed using proteinase K (100 μg ml $^{-1}$ in PBS) for 30 min at room temperature. For Igfbp5, heat-induced epitope retrieval with an alkaline buffer (10 mM Tris, 1 mM EDTA, 0.05% Tween-20, pH 9.0) was used. Endogenous peroxidase activity was blocked by 3% $H_2O_2$. Staining was performed using anti-Ezh1 (abcam, ab13665, 1:100), anti-Ezh2 (Millipore, 07-689, 1:1,000), anti-H3K27me3 (Millipore, 07-449, 1:1,000), anti-H3K27me2 (Millipore, 07-452, 1:1,000), anti-collagen X (abcam, ab58632, 1:1,000), anti-Cdkn2a (abcam, ab54210. 1:1,000), anti-Cdkn2b (Origene, TA312926, 1:100), anti-Igfbp3 (LifeSpan BioSciences, LS-C407922, 1:200) and anti-Igfbp5 (Thermo Scientific, PA5-37369, 1:50) antibodies, with a VECTASTAIN ABC kit (Vector Laboratories, Burlingame, CA) followed by the DAB Substrate kit (Vector Laboratories) according to the manufacturer's instructions. Sections were mounted without counterstaining or counter-stained with methyl green.

***In situ* hybridization.** *In situ* hybridization was performed as described previously[39]. Briefly, riboprobes were generated by PCR using mouse growth plates cDNA as template and with primers that contained SP6 promoter (primer sequence for probe provided upon request). Single stranded digoxigenin-labelled riboprobes for in situ hybridization were transcribed using the DIG RNA Labelling Kit (Roche Diagnostics) following the manufacturer's protocol. Riboprobes were purified by Micro Bio-Spin Columns P-30 Tris RNase free (Bio-Rad), followed by alkaline hydrolysis for 30 min. Paraffin-embedded sections of epiphyseal cartilage from 1-week-old mice were hybridized to digoxigenin-labelled riboprobes. For detection, tissue sections were incubated with anti-digoxigenin alkaline phosphatase Fab fragments (Roche) for 2 h at room temperature and treated with NBT/BCIP (Sigma) in the dark until a colorimetric change was detected. Sections were counter-stained with 10% eosin and visualized using a ScanScope CS digital scanner (Aperio Technologies, Inc) under bright-field microscopy.

**Chondrocyte isolation and culture.** Proximal tibias and distal femurs were dissected from 1-week-old mice aseptically and digested in 0.3% collagenase type I (Sigma-Aldrich) in DMEM/F12 medium. The released cells were resuspended and cultured in DMEM/F12 medium (Invitrogen) supplemented with 10% fetal bovine serum (FBS), 1% penicillin (100 U ml $^{-1}$)/streptomycin (100 μg ml $^{-1}$), 50 μg ml $^{-1}$ ascorbic acid in a humidified incubator at 37 °C, 5% $CO_2$. For treatment with Ezh1/2 inhibitor, dimethylsulphoxide (DMSO) or UNC1999 (2 μM final concentration, Sigma-Aldrich) were added to culture medium 16 h after plating.

**Proliferation assessment by $^3$H-thymidine uptake.** Twenty-four hours after $2 \times 10^5$ chondrocytes were plated per well in 12-well plates, cells were incubated with 1 ml of fresh culture medium containing 1 μCi of $^3$H-thymidine (63 Ci mmol $^{-1}$, MP Biomedicals, Santa Ana, CA) for 16 h, and then vigorously washed three times with PBS. Chondrocytes were detached by collagenase digestion (0.3%, 30 min) and radioactivity was measured by liquid scintillation counting.

**Transfection of chondrocytes with siRNA.** Forty hours after $1 \times 10^5$ chondrocytes were plated per well in 12-well plates, hyaluronidase (5 U ml $^{-1}$, Sigma-Aldrich) was added to the cultured cells for 6 h. One hour before transfection, cells were washed once in PBS and changed to DMEM/F12 medium without antibiotics. Negative control siRNA or siRNA against Cdkn2a (Life Technologies, s63820) and/or Cdkn2b (Life Technologies, s63823) were transfected (40 pmol per reaction) into chondrocytes using Lipofectamine 2000 (Life Technologies) following the manufacturer's standard protocol. Cells were changed back to regular culture medium 16 h post transfection. For treatment with Ezh1/2 inhibitor, DMSO or UNC1999 (2 μM final concentration, Sigma-Aldrich) was also added to culture medium 16 h post transfection, and proliferation or RNA extraction was performed a further 24 h afterwards.

**Chondrocyte pellet culture.** Primary chondrocytes isolated from 1-week-old mice as described above were centrifuged in a 15 ml Falcon polypropylene conical tube (Corning, Tewksbury MA) for 10 min at 400 g to form chondrocyte pellets ($2.4 \times 10^5$ chondrocytes per pellet). Pellets were cultured in a humidified incubator at 37 °C, 5% $CO_2$. In the first 16 h, pellets were maintained in 500 μl regular culture medium (with 10% FBS), before switching to 500 μl culture medium with reduced FBS (DMEM/F12, 0.1% FBS, 1% Pen/Strep, 50 μg ml $^{-1}$ ascorbic acid) afterward. Medium was refreshed every other day. For treatment with Ezh1/2 inhibitor, DMSO or UNC1999 (2 μM final concentration, Sigma-Aldrich) was also added 16 h after pellet formation, and refreshed every other day.

**Metatarsal culture**. Mouse metatarsal culture was performed as previously described[20]. Briefly, the middle three metatarsals were aseptically dissected from wild-type C57BL/6 mice. Bones were maintained in 500 μl α-MEM medium supplemented with 0.2% bovine serum albumin, 0.1 mM β-glycerophosphate, 50 μg ml$^{-1}$ ascorbic acid, 1% Pen/Strep and 0.1% Fungizone. Bone were kept individually in 24-well plates in a humidified incubator at 37 °C, 5% $CO_2$, for 3 days, medium was refreshed on day 2 of culture. DMSO, UNC1999 (2 μM), recombinant mouse Igf1 (100 ng ml$^{-1}$, R&D systems, Minneapolis, MN), recombinant mouse Igfbp3 (50 nM, R&D systems) and/or recombinant mouse Igfbp5 (50 nM, R&D systems) were added at the beginning of culture and refreshed on day 2. Hypertrophic cell size in metatarsal bones was determined taking an area of 100 μm × 100 μm at the centre of the HZ, and measuring and averaging the height of every cell within the area.

**Western blot**. Protein was isolated from cultured chondrocytes using RIPA buffer supplemented with proteinase inhibitor cocktail (Sigma-Aldrich). Western blotting was performed as previously described[40], using anti-Cdkn2a (abcam, ab189034, 1:1,000), anti-Cdkn2b (Origene, TA312926, 1:1,000) and anti-Gapdh (abcam, ab9485, 1:5,000).

**RNA extraction and purification**. For monolayer chondrocytes, RNA was extracted using an RNeasy Mini kit (Qiagen, Valencia, CA). For LCM-dissected growth plate samples, RNA was extracted using a PicoPure RNA Isolation kit (Applied Biosystems). For chondrocyte pellets and metatarsal growth plates, RNA was extracted using a guanidinium isothiocyanate solution, followed by proteinase K digestion and phenol/chloroform extraction, as previously described[41]. All RNA samples had a 260/280 nm ratio between 1.8 and 2.1. RNA integrity was determined using an Agilent 2100 Bioanalyzer (Agilent Technologies, Santa Clara, CA) and only high-quality RNA (RIN > 8) was used for real-time reverse transcription–PCR.

**Quantitative real-time reverse transcription–PCR**. Real-time PCR was used to assess messenger RNA levels in different zones of the growth plate and chondrocytes in different conditions. Total RNA (50–500 ng) was reverse-transcribed using SuperScript IV Reverse Transcriptase (Invitrogen). Quantitative real-time PCR was performed as previously described[42].

**Chromatin immunoprecipitation assay**. ChIP was performed in mouse chondrocytes as previously described[42] with the following modifications; genomic DNA shearing was performed using the Covaris TruCHIP non-ionic Chromatin Shearing kit (Covaris Inc, Woburn, MA) and sonication was performed using the Covaris S2 Ultrasonic System (Covaris Inc). Rabbit polyclonal antibodies against H3K27me3 (Millipore, 07-449) or IgG (Millipore, PP64) as a negative control were used for immunoprecipitation. After ChIP, the recovered DNA was analysed by real-time PCR using primers specific to regions close to transcription start sites of Cdkn2a, Cdkn2b, Igfbp3, Igfbp5 and Gapdh as a negative control, and normalized to the starting DNA (Input). The following primer sequences were used: Cdkn2a, 5′-GTCCGATCCTTTAGCGCTGTT-3′, 5′-AGCCCGGACTACAGAAGAGATG-3′; Cdkn2b, 5′-CGACGGGAGGCAGGTTTT-3′, 5′-CAATCTAGTGCCGAGG GATGTT-3′; Igfbp3, 5′-GCAGATGCGTCCGCTAAAA-3′, 5′-TACGTCGTTGCA CTTTCCCAG-3′; Igfbp5, 5′-CTTTCCGTACATTCCGTGAGG-3′, 5′-AAGGA TGCCAAGGAGCTTTG-3′; Gapdh, 5′-CATCTTCTTGTGCAGTGCCAG-3′, 5′- AGCATCCCTAGACCCGTACAGT-3′; Collagen X, 5′-CTTGCTAGCCCCAAGA CACAA-3′, 5′-CCATGCATCATTCCGCTGTAC-3′; and Ihh, 5′-TCCTGACAA GGCTACATTGCC-3′, 5′-ACCCGAAAGCTGGAGGTCATT-3′.

**Microarray analysis**. For each animal, total RNA from the PZ or HZ (n = 6 each) was amplified using the WT-Ovation Pico RNA Amplification System (NuGEN, San Carlos, CA) following the manufacturer's protocol, and hybridized to Affymetrix Mouse Gene 2.0 ST Array (Affymetrix). Each array represents a single zone from a single animal and a total of six wild-type and six Ezh1/2 mice were used (three males and three females each, all at 3 days old). Affymetrix CEL files were imported into Partek Genomics Suite 6.6 (Partek Inc., St Louis, MO) using the Partek default normalization parameters. Probe-level data were pre-processed, including background correction, normalization and summarization, using robust multi-array average analysis, and log2-transformed. Canonical signalling pathway analysis was performed using Ingenuity Pathways Analysis Software (Ingenuity Systems Inc., Redwood City, CA) and heatmap with two-way hierarchical clustering were constructed using JMP 11 software (SAS Institute Inc., Cary, NC).

**Statistical analyses**. Data are presented as mean ± s.e.m. All statistical analyses were performed in SigmaPlot 11. One-way analysis of variance was used when measuring the effect of a single factor, such as the effect of UNC treatment. Two-way analysis of variance was used when the effect of more than one factors were being assessed. P values were corrected for multiple comparisons, whenever applicable, using the Holm–Sidak method.

**Data availability**. Micoarray data that support the findings of this study have been deposited in GEO with the primary accession code GSE84198. The data that support the findings of this study are available from the corresponding author on request.

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

## Acknowledgements

We thank Jeffrey Hanson from LCM core of NCI for advice and guidance of micro-dissection of growth plate zones with LCM equipment. We thank Dr Yoshiyuki Wakabayashi from DNA Sequencing and Computational Biology Core of NHLBI for help in chromatin shearing and sonication of chondrocytes. We thank Dr Ge Kai from NIDDK for useful advice on Ezh2. This work was supported by the Intramural Research Program of the Eunice Kennedy Shriver National Institute of Child Health and Human Development, NIH. In addition, O.N. was supported by grants from the Swedish Research Council (grant nos 521-2014-3063 and 2015-02227), the Swedish Governmental Agency for Innovation Systems (Vinnova; 2014-01438), Marianne and Marcus Wallenberg Foundation, the Stockholm County Council, Byggmästare Olle Engkvist's Foundation, Stiftelsen Frimurare Barnhuset i Stockholm and Karolinska Institutet.

## Author contributions

J.C.L., O.N. and J.B. conceived the project. J.C.L. and J.B. designed the experiments. J.C.L., P.G., Q.N., M.A., C.K., W.C., Y.H.J., E.L., O.N. and K.M.B. performed the experiments, J.C.L. and J.B. analysed the data and wrote the manuscript.

## Additional information

**Competing financial interests:** The authors declare no competing financial interests.

