## [Peer Review File · Nature Communications]

Reviewers' comments:

Reviewer #1

(Remarks to the Author):

This is a very interesting study that aims to clarify the mechanisms by which histone methyltransferases EZH1 and EZH2 regulate chondrocyte activity within the growth plate. The information gleaned will help us understand better the epigenetic control of endochondral ossification and specifically inform on the molecular and cellular mechanisms underpinning the skeletal overgrowth observed in Weaver syndrome. Using their knowledge of the literature and chondrocyte function the authors have used a multi-approach strategy and focussed on two mechanisms that may explain the altered chondrocyte proliferation and differentiation that was noted in Ezh1/2 KO mice. The authors conclude that H3K27 methylation promotes chondrocyte proliferation by repressing cdk_i Ink4a/b activity whereas H3K27 control of hypertrophic enlargement is via down regulation of igfbp3&5 expression and elevated IGF signalling (a recognised +ve stimulator of chondrocyte hypertrophy). Hence mutations in EZH1 and EZH2 result in less H3K27 methylation and unrepressed cdk_i Ink4a/b and igfbp 3&5 expression. Both decreased chondrocyte proliferation and hypertrophy combine together to explain the reduced longitudinal bone growth observed in the Ezh1/2 KO mice (the authors do recognise that these observations are not consistent with the skeletal over growth observed in Weavers syndrome). Whilst the authors elegant studies make a strong case for these cellular mechanisms to be at the centre of the delayed endochondral ossification in Ezh1/2 mice it is likely that inhibiting H3K27 transcriptional repression ability will have many wide ranging effects and would not be limited to derepression of only igfbp and Cdkn2a/2b expression. More global fishing type studies - microarray, NGS may be required to identify the full repertoire of changes in chondrocyte gene expression that is a consequence of the loss of histone methyltransferases Ezh1 and Ezh2.

Other Major points.

1. As endochondral ossification is required for linear bone growth during embryonic and postnatal life can the authors speculate why the body weight and tibial lengths of the Ezh1/2 mice look apparently normal up to about 3/4 days of post-natal growth (Figs 1b and 2i)? In Fig 1b the 1 week old KO pup looks very growth retarded whereas the body measurement in Fig 1b looks only slightly done. This looks a mismatch.
2. Fig 1c-h. Some higher power magnifications would be useful to localize Ezh2 and H3K27 to the chondrocytes. Cannot see much at this magnification. What about Ezh1?. Also cannot see much Ezh2 in the primary spongiosa of the Ezh1/2 mice. This should be present in osteoblasts and other cells and higher magnifications would help.
3. Fig 2i & j. Some indication of statistical significance would be useful (also fig 3d, 3g4k & n - maybe others). It is interesting that at 14 days PZ height and cells/column were normal in the Ezh1/2 mice - is there any obvious explanation for this. Is there any evidence in older mice (beyond P14) that linear bone growth normalizes?
4. Fig. 3f - Using LCM and qPCR the authors show significantly reduced Ezh2 in the growth plate of the KO mice but expression was still measureable. Is this expression consistent with the IHC results in Fig 1f?
5. The data with the growth plates, isolated cells, UNC use and ChIP assays together convincingly show that alterations in H3K27 methylation result in increased Cdkn2a/2b which is associated with decreased chondrocyte proliferation. The data supports the ex-vivo data. However when the authors went out to confirm that the decrease in proliferation is driven by elevated Cdkn2a/2b levels the story comes a little unstuck. The siRNA data is not that supportive. The rescue of chondrocyte proliferation was minimal - only by about 10-15% and there was still a greater the 25% reduction in proliferation in the presence of siRNA to Cdkn2a/2b. These siRNA data suggest that this may not be the major mechanism for the reduced proliferation.

6. Could an additional explanation for the decreased chondrocyte proliferation in the Ezh1/2 mice be the increased IGF binding proteins (noted in the HZ)? IGF-1 is a recognised mitogen and it is likely if IGFBPs are raised in the proliferating chondrocytes of the Ezh1/2 mice this would result in diminished IGF mitogenic activity due to diminished bioavailability. It is hard to think why only HZ chondrocytes would have increased IGFBP in the absence of ezh1/2 so the authors should have also looked for IGFBP3 and 5 expression in the PZ chondrocytes of the growth plate? In Fig 4A why was this analyses limited to HZ chondrocytes? Further expts could include adding IGFBP to the metatarsal cultures in a similar way as described in fig 4n-p and measuring chondrocyte proliferation rather than hypertrophy.
7. Can the authors explain why the regulation of igfbp3 and 5 expression by UNC is different between pellet and monolayer cultures (Figs 4c and 4g)? The increase in igfbp5 in the pellet cultures is also not that convincing. It is only elevated at 1 time point (2-days). Would this be sufficient to alter IGF availability to the cells.
8. To directly examine if the increased igfbp3 and 5 is the cause of reduced hypertrophy a similar strategy used to study Cdkn2a/2b role in proliferation should be employed. If the authors are correct the knockdown of igfbp3&5 by siRNA should rescue the reduction hypertrophic chondrocyte size by UNC treatment.
9. The increase in metatarsal growth with IGF-1 is not likely to be solely through their positive actions on chondrocyte hypertrophy (Fig. 4k-m). Even in the presence of increased Cdkn2a/2b, IGF-1 may be able to promote chondrocyte proliferation. Also the addition of IGFBP3&5 will reduce IGF-1 ability to promote proliferation and bone length (Fig. 4n-p). Therefore the changes in bone length with IGF-1 and IGFBP addition are unlikely only to be explained by changes in chondrocyte hypertrophy. Maybe this should be made clear.
10. IHC to show changes in Cdkn2a/2b and igfbp3&5 protein expression to specific parts of the growth plate of Ezh2 mice would complement the LCM/gene expression data.

Minor Points

1. Line 57: "duplication of "invaded by"
2. Line 70: space after PRC2
3. Line 96: double full stop
4. Worth saying why conditional Ezh2 mice were used - I assume global knockouts were lethal?
5. Cannot see any mention of gender of mice studied.
6. What was the purpose of studying Col2-cre Ezh2+/fl. Is there not a case for studying Ezh2fl/fl (without Cre) to rule out a bone phenotype when floxing the Ezh2 gene.
7. Line 94: Linear growth was not measured in fig 1b - only body weight.
8. Line 99: take out "the".
9. Page 108/109: Ezh1/2 already defined - no need to do it again.
10. Fig 2. I assume the growth plate images of the Col2-cre Ezh2+/fl mice were very similar to the control mice and this is the reason they have not been shown. If this is indeed the case then this should be stated.
11. When the authors say they measured the size of the hypertrophic chondrocyte do they actually mean the size of the lacunae? The actual chondrocytes from the images look rather shrunken. If this is indeed the case then the text should be modified.
12. The y-axis in figs 4c,g,I should be labelled.
13. Figs 4h and 4k and n. The latter 2 figs show a dramatic reduction in the length of metatarsals incubated with UNC (approx. 50%). This reduction in growth is not obvious in Fig 4h. Is this a representative image?

Reviewer #2 (Remarks to the Author):

The authors use a conditional knockout model (Col2-Cre) to ablate the Ezh2 gene in developing

cartilage (on a Ezh1 null background). Several stage-specific defects in chondrocyte maturation were observed and attributed to the misexpression of Cdkn2a/b and Igf-family proteins in proliferating and hypertrophic chondrocytes, respectively.

General comments:

The manuscript presents an extensive assessment of Ezh1/2 deletion in chondrocytes (via Col2-Cre). There are no real concerns with the methodology or presentation of the findings, however the relative novelty and impact of the findings are questionable for a journal of this caliber. Several of the results described are similar to an earlier study describing the knockout of Ezh2 via Prrx-Cre. (Dudakovic A et al. Epigenetic Control of Skeletal Development by the Histone Methyltransferase Ezh2. J Biol Chem. 2015 Nov 13;290(46):27604-17. PMID:264247900. Although Dudakovic et al primarily examine gene expression changes related to osteogenesis/MSC differentiation, there is extensive description of changes related to growth plate abnormalities, chondrogenesis in that manuscript. At a minimum this paper (Dudakovic et al) should be referenced and discussed.

Although the authors state "... (encoded by Cdkn2a and 2b) inhibits cell proliferation and is a recognized molecular target of the PRC2 complex. We therefore hypothesized that loss of Ezh1 and 2 allows derepression of Cdkn2a..." (PG5 LN 129), it is not clear why Cdkn2a is the sole target of Ezh1/2 studied. The PRC2 complex targets several hundred genes, many of which would presumably influence chondrocyte proliferation. The coordinated and reciprocal expression of Cdkn2a and Ezh2 (both in knockout and inhibition models) would suggest that Cdkn2a is involved in the decreased chondrocyte proliferation however, there are most likely many other direct and indirect gene expression changes in these cells. Presumably since Ezh1/2-containing complexes regulate most H3K27me3 in these cells, knockdown of Ezh2 would reduce H3K27me3-mediated repression globally within these cells (somewhat supported by Figure 1). A similar critique could be suggested for the effects of Igfr/Igfbp genes in the hypertrophic zone (figure 4). Although it would be a large amount of additional work, a full characterization of these expression and/or H3K27me3 changes should be performed given the relatively nonspecific action of Ezh2 in developmental systems/gene expression.

Although the manuscript constitutes a large amount of excellent work, for the previously outlined reason the manuscript can not be recommended for publication in the current form.

Specific Comments

[Figure 2] It is unclear from the images in panel g/h that hypertrophic chondrocyte size is altered in KO animals versus controls. Is this measurement statistically significant?

[Figure 3] Knockdown experiment of Cdkn2a/b should be accompanied by a Western blot to determine relative protein levels upon knockdown. Additionally, it is not clear why Gapdh, a gene presumably not repressed/marked by H3K27me3 was used as a control in this experiment. What exactly does this control for? IgG is a sufficient negative control. A lineage restricted gene would serve as a much better indicator in this experiment as it should maintain repression during chondrogenesis.

Recommendation: Major revision

Reviewer #3 (Remarks to the Author):

Summary

This study reports at a very novel and significant finding demonstrating that the contribution of the Ezh enzymes to chondrogenesis. This is an important paper for several reasons, 1.) It's well known that only this enzyme family produces the repressive histone modification mark H3K27me3, and 2.) Knockdown of either Ezh1 or Ezh2 alone has no effect on skeletal development; however, the double

KO (Col2 cartilage-specific cre Ezh1^{-/-}-Ezh2 fl/fl) has severely impaired overall growth of the mouse. A continuing decrease in the columnar proliferation zone from 3 to 14 days age is observed and consequently, the same proportional decrease in the pre hypertrophic zone. Mechanisms contributing to decelerated maturation of chondrocytes forming the growth plate zones and thus endochondral bone formation (EBF) were established.

The authors demonstrated in a series of rigorous studies using several approaches a proliferation defect, an increase in H3K27me3 and increased Cdkn2a, which is target of the PRC2 complex that includes EZH2. Confirmation of this mechanism was validated by analyses of the gene in isolated cartilage zones (by LCM) and by ex vivo chondrocyte culture treated with Ezh1/2 inhibitors. The authors then examined deregulated IGF signaling contributing to the diminished HZ, as IGF impaired signaling would account for retarded overall growth. Again rigorous studies identified two highly upregulated IGF inhibitor binding proteins (BP3, BP5) with no change in expression of IGF or IGF1R. Strong evidence is presented including rescue of the phenotype. In vitro and in vivo.

Suggested Improvements:

While there is enthusiasm for this study, and the mice will surely be wanted numerous laboratories, there are still questions that can be readily addressed.

- 1.) Although the mechanism for inhibited proliferation and maturation of the HZ are validate, there are indications from the histological sections that more is going on. Two of the critical regulatory factors distinctly expressed in each zone that regulate the timing of chondrocyte maturation for EBF, namely PTHrP and IHH, are involved in intricate feed -back and forward signaling that control differentiation. Type X collagen and IHH are decreased in cultured chondrocytes (Fig 4g), but perhaps an immunohistochemical staining would better identify the molecular signature of the in vivo HZ cells. The fact that the DKO HZ cells are smaller at day 14 suggests they are the IHH signaling cells. Performing an IHC and also Type X IHC would clarify the block in differentiation.
- 2.) The source of the lack of clarity can be solved by showing images of the full length bone, for Fig 2 and Fig 4 particularly panel L and O. (these can be in supplement) as a supplement.
- 3.) Figure 2- the growth plate zones should be labeled on the panels: RZ for resting zone, PZ proliferating zone and more accurate labeling of the hypertrophic zones for, HZ (mature, hypertrophic cells cartilage not calcified) and terminal HZ (calcified cartilage). There is also a Pre -HZ that is not easy to see at day 7, but on day 14 is evident only in the WT mouse and extends a 2-3 mm below the lower dotted line. In striking contrast, that transition is not present in DKO mice, but clearly shows you have Hypertrophic Zone of uncalcified cartilage and a terminal HZ calcified cartilage zone which is Mallory Trichrome stained and weakly in DKO bone.
- 4.) Size concerns. Figure 1 shows mouse half the size at 1 wk, but growth chart very close at 1 week. Figure 1i lower limbs - are missing? Were these removed for analyses or are they not developed at day 3. Figure 2 at 14 day shows an epiphysis with an articular surface and a little bit of joint. There is no secondary center of ossification. In full sections of the limb did the ephysis change in proportion to bone size. Again having full length sections shown so that the primary spongiosa can be visualized would be informative for understanding the phenotype especially after treatment with IGF
- 5.) A ChIP was performed with H3K27 me 3. This ChIP could have been analyzed for ColX and IHH - as nothing is known related to their repression by epigenetic regulators.

Discussion and References:

With only 4 papers in the literatures related to the author's studies, they could have been indicated.

Particularly worthy of comparison in Discussion is the cartilage phenotype resulting from homozygous deletion of K27me3 by the demethylase JMJD3.

Reviewer #1

Reviewer comment: This is a very interesting study that aims to clarify the mechanisms by which histone methyltransferases EZH1 and EZH2 regulate chondrocyte activity within the growth plate. The information gleaned will help us understand better the epigenetic control of endochondral ossification and specifically inform on the molecular and cellular mechanisms underpinning the skeletal overgrowth observed in Weaver syndrome. Using their knowledge of the literature and chondrocyte function the authors have used a multi-approach strategy and focussed on two mechanisms that may explain the altered chondrocyte proliferation and differentiation that was noted in Ezh1/2 KO mice. The authors conclude that H3K27 methylation promotes chondrocyte proliferation by repressing cdki Ink4a/b activity whereas H3K27 control of hypertrophic enlargement is via down regulation of igfbp3&5 expression and elevated IGF signalling (a recognised +ve stimulator of chondrocyte hypertrophy). Hence mutations in EZH1 and

EZH2 result in less H3K27 methylation and unrepressed cdki Ink4a/b and igfbp 3&5 expression. Both decreased chondrocyte proliferation and hypertrophy combine together to explain the reduced longitudinal bone growth observe in the Ezh1/2 KO mice (the authors do recognise that these observations are not consistent with the skeletal over growth observed in Weavers syndrome). Whilst the authors elegant studies make a strong case for these cellular mechanisms to be at the centre of the delayed endochondral ossification in Ezh1/2 mice it is likely that inhibiting H3K27 transcriptional repression ability will have many wide ranging effects and would not be limited to derepression of only igfbp and Cdkn2a/2b expression. More global fishing type studies - microarray, NGS may be required to identify the full repertoire of changes in chondrocyte gene expression that is a consequence of the loss of histone methyltransferases Ezh1 and Ezh2.

Response: The reviewer makes an important point. While we have shown evidence that Cdkn2a/b and Igfbp3/5 are important pathways mediating the effects of H3K27 methylation on growth plate chondrogenesis, it is likely that de-repression of other genes also contributes. As the reviewer suggested, we performed expression microarray as a preliminary exploration of these other possible pathways. To describe the findings, we have added the following paragraph near the end of the Results section:

“To explore other potential pathways by which loss of Ezh1 and 2 may affect growth plate chondrogenesis, we used microarray to analyze differences in gene expression in the growth plate between Ezh1/2 and wild-type mice. We used laser capture microdissection to isolate growth plate tissue from either proliferative or hypertrophic zone, isolated RNA, and performed microarray analysis and pathway analysis. We found that 49 genes were downregulated ($P < 0.001$, fold-change > 1.3) in PZ of Ezh1/2 mice while 231 genes were upregulated compared to wild type. Similarly in HZ, 97 genes were downregulated ($P < 0.001$, fold-change > 1.3) and 146 genes were upregulated in the Ezh1/2 mice, compared to the wild type mice (Fig 5a, c). To confirm the validity of the microarray analysis, we performed real-time PCR on a subset of genes that showed differential regulation (Fig. 5e). The observation that more genes were upregulated in Ezh1/2 mice (as opposed to downregulated), both in PZ and HZ, is consistent with the role of Ezh1 and 2 in catalyzing H3K27 trimethylation, which serves as a repressive mark of gene

expression. Interestingly, the number of genes that were upregulated or downregulated in both the PZ and the HZ was modest (Fig. 5b). Thus the targets of H3K27 repression appears to depend on the differentiation state of the chondrocytes. The microarrays results also confirmed the upregulation of Cdkn2a/b in PZ and the upregulation of Igfbp3/5 in HZ (supplemental data). Next, we used Ingenuity Pathway Analysis to explore additional potential signaling pathways (Fig. 5d). As expected, this analysis implicated cell-cycle G1/S regulation in the PZ (which includes upregulation of Cdkn2a/b) and implicated IGF signaling in HZ (which includes upregulation of Igfbp3/5). In addition, the analysis implicated Wnt/beta-catenin and glucocorticoid receptor signaling in PZ and thyroid hormone receptor and retinoid (RAR) signaling, all of which are thought to play important roles in the regulation of growth plate chondrocytes.”

Other Major points:

Reviewer comment 1: As endochondral ossification is required for linear bone growth during embryonic and postnatal life can the authors speculate why the body weight and tibial lengths of the Ezh1/2 mice look apparently normal up to about 3/4 days of post-natal growth (Figs 1b and 2i)?

Response: As the reviewer notes, the growth of the knockout mice appears normal during prenatal development and then becomes abnormal postnatally so that a noticeable difference in body size becomes evident by about 3 days after birth. We speculate that this timing in some way reflects the major differences in the structure and function of the prenatal and postnatal growth plate cartilage. For example, in the embryonic cartilage, the epiphyseal chondrocytes proliferate actively, contributing to growth, whereas, in the postnatal cartilage, the epiphyseal chondrocytes are partially replaced by bone and the remaining chondrocytes become far more quiescent as resting zone chondrocytes. However, we do not know why Ezh1/2 affects primarily the postnatal and not prenatal form of endochondral bone formation. We therefore modified the manuscript to read:

“Unlike the ubiquitous knockout of Ezh2, which is embryonic lethal²³, mice lacking both Ezh1 and Ezh2 in the cartilage (Col2-cre Ezh1^{-/-} Ezh2^{fl/fl}, hereafter termed Ezh1/2 mice) were born at decreased frequency compared to the expected Mendelian ratio of 25% (Table S1), but appeared normal in size at birth. However, by 3 days after birth, the Ezh1/2 mice were noticeably smaller in size than wild-type mice, and the difference became more prominent with age (Fig. 1a,b). We speculate that this greater effect on skeletal growth after birth in some way reflects the major differences in the structure and function of the prenatal and postnatal growth plate cartilage²².”

Reviewer comment 1 (continued): In Fig 1b the 1 week old KO pup looks very growth retarded whereas the body measurement in Fig 1b looks only slightly done. This looks a mismatch.

Response: As the reviewer also points out, the difference in body size of the mice in Fig 1b appears greater than the average difference in body weight shown in Fig 1b or the difference in tibial length shown in Fig. 2i. We have therefore bred more mice and photographed a pair of mice that more closely approximates the average difference in size at this age.

Reviewer comment 2: Fig 1c-h. Some higher power magnifications would be useful to localize

Ezh2 and H3K27 to the chondrocytes. Cannot see much at this magnification. What about Ezh1? Also cannot see much Ezh2 in the primary spongiosa of the Ezh1/2 mice. This should be present in osteoblasts and other cells and higher magnifications would help.

Response: We have now included higher magnifications as the reviewer requested (Fig 1c-h). As the reviewer predicted, Ezh2 continues to be expressed in osteoblasts of the spongiosa because these cells do not express Col2-Cre. In response to the reviewer's question, we also performed Ezh1 immunostaining on the mouse with homozygous knockout of Ezh1 and found that the protein was absent in all cells examined, whereas Ezh1 was well expressed in the heterozygous mouse (Supplemental Fig S3).

Reviewer comment 3: Fig 2i & j. Some indication of statistical significance would be useful (also fig 3d, 3g4k & n - maybe others).

Response: As requested, we have added P values to indicate statistical significance in figs 2, 3, and 4.

Reviewer comment 3 (continued): It is interesting that at 14 days PZ height and cells/column were normal in the Ezh1/2 mice - is there any obvious explanation for this. Is there any evidence in older mice (beyond P14) that linear bone growth normalizes?

Response: The findings at 14 days are more difficult to interpret because, by that time, the mice showed evidence of respiratory insufficiency, which is likely due to insufficient growth of the rib cage and by the abnormal the vertebral column. For that reason, we focused our studies on the first week of life, when there is no evidence of respiratory insufficiency and therefore the effects on the growth plate are likely directly caused by the genetic defects.

Reviewer comment 4: Fig. 3f - Using LCM and qPCR the authors show significantly reduced Ezh2 in the growth plate of the KO mice but expression was still measureable. Is this expression consistent with the IHC results in Fig 1f?

Response: The immunohistochemistry did appear to show slight residual staining for Ezh2 in the growth plate of Ezh1/2 mice, which is consistent with the qPCR. We hope the high magnification inset demonstrates this finding better.

Reviewer comment 5: The data with the growth plates, isolated cells, UNC use and ChIP assays together convincingly show that alterations in H3K27 methylation result in increased Cdkn2a/2b which is associated with decreased chondrocyte proliferation. The data supports the ex-vivo data. However when the authors went out to confirm that the decrease in proliferation is driven by elevated Cdkn2a/2b levels the story comes a little unstuck. The siRNA data is not that supportive. The rescue of chondrocyte proliferation was minimal - only by about 10-15% and there was still a greater the 25% reduction in proliferation in the presence of siRNA to Cdkn2a/2b. These siRNA data suggest that this may not be the major mechanism for the reduced proliferation.

Response: We agree with the reviewer that the siRNA findings support the hypothesis that increased Cdkn2a/b expression contributes to the impaired proliferation but also suggests that the increase in Cdkn2a/b is not the only mechanism and thus other pathways are also likely to play important roles. Based on the reviewer's comment, we have tried to convey this conclusion more clearly in the manuscript:

“In contrast, treatment of UNC1999 led to suppression of chondrocyte proliferation, which was then partially reversed by siRNA against Cdkn2a/b (Fig. 3i). Our data therefore suggest that the decrease in chondrocyte proliferation induced by loss or inhibition of Ezh1/2 is partially attributable to derepression of Cdkn2a/b expression, but that other molecular pathways may also make important contributions to the impaired proliferation.”

Reviewer comment 6: Could an additional explanation for the decreased chondrocyte proliferation in the Ezh1/2 mice be the increased IGF binding proteins (noted in the HZ)? IGF-1 is a recognized mitogen and it is likely if IGFBPs are raised in the proliferating chondrocytes of the Ezh1/2 mice this would result in diminished IGF mitogenic activity due to diminished bioavailability. It is hard to think why only HZ chondrocytes would have increased IGFBP in the absence of ezh1/2 so the authors should have also looked for IGFBP3 and 5 expression in the PZ chondrocytes of the growth plate? In Fig 4A why was this analyses limited to HZ chondrocytes? Further expts could include adding IGFBP to the metatarsal cultures in a similar way as described in fig 4n-p and measuring chondrocyte proliferation rather than hypertrophy.

Response: The reviewer raises an interesting hypothesis that the increase in Igfbp3 and 5 might contribute not only to the diminished hypertrophy but also to the diminished proliferation. However, our real-time PCR (Fig.4a) and microarray data (Fig.5 and supplemental data) showed that Igfbp3/5 were only elevated in the HZ of Ezh1/2 mice but not well-expressed in the PZ, either in the wild-type or the Ezh1/2 mice. To address this point, we amended the results as follows:

“However, we found that two members of the Igf binding protein (Igfbp) family, Igfbp3 and 5, were significantly upregulated specifically in the HZ of the Ezh1/2 mice compared to wildtype littermates (Fig. 4a).”

Reviewer comment 7: Can the authors explain why the regulation of igfbp3 and 5 expression by UNC is different between pellet and monolayer cultures (Figs 4c and 4g)? The increase in igfbp5 in the pellet cultures is also not that convincing. It is only elevated at 1 time point (2-days). Would this be sufficient to alter IGF availability to the cells.

Response: Although the increase in Igfbp5 in the pellet culture only reaches statistical significance at one time point in post hoc pairwise comparisons, the overall effect of treatment, on Igfbp5 mRNA levels across all time points was statistically significant ($P < 0.006$), consistent (upregulated in all time points examined), and reasonable in magnitude (upregulated 1.5-2-fold at each time point), a difference which, we think, could have biological effects. We agree with the reviewer's observation that the regulation of Igfbp3 and 5 appears to be different in the monolayer and pellet cultures, with Igfbp3 upregulated in monolayer and Igfbp5 upregulated in

pellet culture by UNC treatment. We do not know the specific reasons for the difference. The two models systems differ in that the pellet culture allows greater cell-cell signaling, altered cell-matrix interaction, and allows chondrocyte hypertrophic differentiation; however we do not know how these difference between model systems specifically affects regulation of these two genes. In contrast, in the metatarsal organ culture, which represents the most physiological of our in vitro models, both Igfbp3 and 5 were upregulated by UNC treatment. For this reason, we used the metatarsal system to perform mechanistic studies using exogenous IGF-I and IGFBP3/5. To acknowledge this difference between the monolayer, pellet, and metatarsal systems, we added the following statement to the manuscript:

“Treatment with UNC1999 decreased H3K27me2 and me3 (Fig. 4j, Fig. S10), and increased expression of Igfbp3 and 5 in the metatarsal growth plate (Fig. 4i). This upregulation of both Igfbp3 and 5 is similar to the upregulation of both genes observed in the Ezh1/2 mice, and thus the cultured metatarsal bones may serve as a more physiological model system than either the monolayer culture, in which only Igfbp3 was upregulated or the pellet culture, in which only Igfbp5 was upregulated.”

Reviewer comment 8: To directly examine if the increased igfbp3 and 5 is the cause of reduced hypertrophy a similar strategy used to study Cdkn2a/2b role in proliferation should be employed. If the authors are correct the knockdown of igfbp3&5 by siRNA should rescue the reduction hypertrophic chondrocyte size by UNC treatment.

Response: We agree that this experiment would further confirm the causal role of Igfbps, but unfortunately we do not know of a good model system in which to perform this experiment. In models that allow growth plate chondrocyte hypertrophy, such as pellet culture or metatarsal culture, transfection has not, to our knowledge, been achieved. This practical limitation was part of our motivation for using pharmacological manipulation, rather than using siRNAs. We showed that addition of IGF-I to the culture medium of metatarsal bones overcame the effect of decreased H3K27 methylation on hypertrophy whereas addition of Igfbps to the culture medium reproduced the effect on hypertrophy. We think that these pharmacological experiments serve reasonably well in the place of the siRNA experiments to provide evidence for the proposed molecular pathway.

Reviewer comment 9: The increase in metatarsal growth with IGF-1 is not likely to be solely through their positive actions on chondrocyte hypertrophy (Fig. 4k-m). Even in the presence of increased Cdkn2a/2b, IGF-1 may be able to promote chondrocyte proliferation. Also the addition of IGFBP3&5 will reduce IGF-1 ability to promote proliferation and bone length (Fig. 4n-p). Therefore the changes in bone length with IGF-1 and IGFBP addition are unlikely only to be explained by changes in chondrocyte hypertrophy. Maybe this should be made clear.

Response: As suggested by the reviewer’s comments, we have performed BrdU labeling and staining in metatarsal culture to study the effect of IGF-1 and UNC on metatarsal chondrocyte proliferation (Fig S11). We found that the effect of IGF-1 on metatarsal proliferation was modest in magnitude and did not reach statistical significance. This is consistent with prior studies by other groups [PMID: 14749359] showing IGF-1 promotes proliferation in the perichondrium but not in chondrocytes in the metatarsal culture system. We therefore conclude that the increase in

metatarsal growth with IGF-1 is primarily attributable to actions on hypertrophy. To address this important point, we modified the manuscript to read:

“Consistent with this prediction, Igf1 treatment overrode the effect of UNC1999 in that hypertrophic cell size was similarly increased in metatarsal bones treated with Igf1 alone and treated with both Igf1 and UNC1999 (Fig. 4l-m). In contrast, Igf1 treatment did not significantly affect chondrocyte proliferation (Fig. S11).”

Reviewer comment 10: IHC to show changes in Cdkn2a/2b and igfbp3&5 protein expression to specific parts of the growth plate of Ezh2 mice would complement the LCM/gene expression data.

Response: As suggested by the reviewer, we have performed immunostaining of Cdkn2a/b and Igfbp3/5 in the growth plate to complement and confirm our gene expression data. The findings are now included in the manuscript as Fig S5.

Minor Points:

1. Line 57: "duplication of "invaded by"
2. Line 70: space after PRC2
3. Line 96: double full stop

Response: We thank the reviewer for the careful reading and have corrected these errors.

Minor Points (continued):

4. Worth saying why conditional Ezh2 mice were used - I assume global knockouts were lethal?

Response: As the reviewer surmised, the global knockout is lethal. We have now stated this in the manuscript.

5. Cannot see any mention of gender of mice studied.

Response: We have now stated in the method that a combination of male and female mice were used. Our experiments involved mice in the first weeks of life when males and females show very similar skeletal growth patterns.

6. What was the purpose of studying Col2-cre Ezh2+/fl. Is there not a case for studying Ezh2fl/fl (without Cre) to rule out a bone phenotype when floxing the Ezh2 gene.

Response: As the reviewer points out, comparing Ezh2 fl/fl Col2-Cre mice versus Ezh2 fl/fl (without Cre) would control for an effect of floxing the Ezh2 gene. Although we did not study the Ezh2 fl/fl mice (without Cre) in detail, we did collect weight data on these mice (Ezh1 -/-, Ezh2 fl/fl) and confirmed that their growth were indistinguishable from Ezh1-/- mice or Ezh1-/-,

Ezh2^{+/-} mice, suggesting that floxing the Ezh2 gene had no growth phenotype in the absence of Col2-Cre. The growth data are now including as Fig S1. Our primary comparison was between Ezh1^{-/-}, Ezh2^{fl/fl}, Col2-Cre mice versus Ezh1^{-/-}, Ezh2^{+/-}, Col2-Cre to control for any confounding effect of Col2-Cre expression. We also included heterozygous Ezh1^{-/-}, Ezh2^{+/-}, Col2-Cre mice and found that these mice with a single remaining intact allele of Ezh2 grew similarly to Ezh1^{-/-}, Ezh2^{+/-}, Col2-Cre, indicating that homozygous loss of Ezh2 is required to produce the skeletal phenotype.

7. Line 94: Linear growth was not measured in fig 1b - only body weight.

Response: We have modified the sentence to read:

“Decreased overall body growth (Fig. 1b) and longitudinal bone growth (Fig. 2i and see below) was observed as early as 3 days of age,”

8. Line 99: take out "the".

9. Page 108/109: Ezh1/2 already defined - no need to do it again.

Response: These errors have been corrected. Thank you.

10. Fig 2. I assume the growth plate images of the Col2-cre Ezh2^{+/-} mice were very similar to the control mice and this is the reason they have not been shown. If this is indeed the case then this should be stated.

Response: This has now been stated:

“The growth plate histology of heterozygotes were indistinguishable from wildtype littermates at the ages studied”

11. When the authors say they measured the size of the hypertrophic chondrocyte do they actually mean the size of the lacunae? The actual chondrocytes from the images look rather shrunken. If this is indeed the case then the text should be modified.

Response: The review makes a valid point. We did measure the size of the lacunae because, as the reviewer observed, the actual hypertrophic chondrocytes condense during tissue fixation/processing. We have stated more clearly in the supplemental online methods section as the reviewer suggested:

“For terminal hypertrophic cell height, the height of the lacunae, which reflects the actual hypertrophic chondrocytes height before cells condense during tissue fixation/processing, were measured.”

12. The y-axis in figs 4c,g,l should be labelled.

Response: We have now labelled the y-axis of these figures.

13. Figs 4h and 4k and n. The latter 2 figs show a dramatic reduction in the length of metatarsals incubated with UNC (approx. 50%). This reduction in growth is not obvious in Fig 4h. Is this a representative image?

Response: The graphs in Fig. 4h and 4k do not represent the absolute length but rather the change in length from the beginning of the experiment. Therefore, a 50% difference in this graph represents a 50% difference in growth rate but not in absolute length. The metatarsals were about 1.8 mm in length at the beginning of the experiment. By the end of the 3-day experiment, as shown in Fig 4k and n, DMSO-treated bones grew to about 2.4 mm (increased by 0.6 mm over the 3 day period), while UNC-treated bones grew to about 2.2 mm (increased by 0.4 mm). As this example illustrates, the effects on the absolute length seen in Fig 4h (2.4 mm vs 2.2mm) were not as dramatic as the effects on growth rate as seen in Fig. 4k and n (0.6 mm vs 0.4 mm).

Reviewer #2

The authors use a conditional knockout model (Col2-Cre) to ablate the Ezh2 gene in developing cartilage (on a Ezh1 null background). Several stage-specific defects in chondrocyte maturation were observed and attributed to the misexpression of Cdkn2a/b and Igf-family proteins in proliferating and hypertrophic chondrocytes, respectively.

General comments:

The manuscript presents an extensive assessment of Ezh1/2 deletion in chondrocytes (via Col2-Cre). There are no real concerns with the methodology or presentation of the findings, however the relative novelty and impact of the findings are questionable for a journal of this caliber. Several of the results described are similar to an earlier study describing the knockout of Ezh2 via Prrx-Cre. (Dudakovic A et al. Epigenetic Control of Skeletal Development by the Histone Methyltransferase Ezh2. J Biol Chem. 2015 Nov 13;290(46):27604-17. PMID:264247900. Although Dudakovic et al primarily examine gene expression changes related to osteogenesis/MSC differentiation, there is extensive description of changes related to growth plate abnormalities, chondrogenesis in that manuscript. At a minimum this paper (Dudakovic et al) should be referenced and discussed.

Response: As suggested, we have added the following to the discussion,

“Several prior studies have investigated the role of Ezh2 and H3K27 methylation in growth plate chondrogenesis. Dudakovic et al. studied a mouse with Ezh2 ablated in uncommitted mesenchymal cells (9). Thus their model differed from ours in that loss of Ezh2, occurred earlier in cell differentiation and Ezh1 was left intact. They observed shortened long bones and vertebrae, reduced hypertrophic zone height and overall growth plate height. Zhang et al studied Jmjd3, which encodes an H3K27me3 demethylase and found that ablation of the gene in mice decreases proliferation and delays hypertrophy of chondrocytes, indicating that increased H3K27

methylation, like the decreased methylation in the current study, can severely impair endochondral bone formation (35). Very recently, Mirzamohammadi et al described a mouse with cartilage-specific loss of Eed, which like Ezh2, is a component of the Polycomb repressive complex 2 that catalyzes trimethylation of H3K27 (22). These mice showed effects similar to Ezh1/2 mice with kyphosis, decreased chondrocyte proliferation, accelerated hypertrophic differentiation and cell death with reduced Hif1a expression. Their study focused on the roles of excessive Wnt and TGF- β signaling.”

The last paper from Mirzamohammadi et al was published in June, 2016, while our study was under review.

General comments (continued):

Although the authors state "... (encoded by Cdkn2a and 2b) inhibits cell proliferation and is a recognized molecular target of the PRC2 complex. We therefore hypothesized that loss of Ezh1 and 2 allows derepression of Cdkn2a..." (PG5 LN 129), it is not clear why Cdkn2a is the sole target of Ezh1/2 studied. The PRC2 complex targets several hundred genes, many of which would presumably influence chondrocyte proliferation. The coordinated and reciprocal expression of Cdkn2a and Ezh2 (both in knockout and inhibition models) would suggest that Cdkn2a is involved in the decreased chondrocyte proliferation however, there are most likely many other direct and indirect gene expression changes in these cells. Presumably since Ezh1/2-containing complexes regulate most H3K27me3 in these cells, knockdown of Ezh2 would reduce H3K27me3-mediated repression globally within these cells (somewhat supported by Figure 1). A similar critique could be suggested for the effects of Igfr/Igfbp genes in the hypertrophic zone (figure 4). Although it would be a large amount of additional work, a full characterization of these expression and/or H3K27me3 changes should be performed given the relatively nonspecific action of Ezh2 in developmental systems/gene expression. Although the manuscript constitutes a large amount of excellent work, for the previously outlined reason the manuscript cannot be recommended for publication in the current form.

Response: We agree with the reviewer’s insight that H3K27 methylation likely regulates gene expression of many genes in the growth plate. Although we have focused on Cdkn2a and 2b and Igfbp3 and 5 and shown evidence that these pathways are important mediators of Ezh1/2 action, there are likely to be other pathways that contribute. To address this question, we have studied changes in expression more globally using microarray. The results is now included as Fig 5 and serve as an exploratory analysis of these other possible pathways. We have added the following paragraph near the end of the Results section:

“To explore other potential pathways by which loss of Ezh1 and 2 may affect growth plate chondrogenesis, we used microarray to analyze differences in gene expression in the growth plate between Ezh1/2 and wild-type mice. We used laser capture microdissection to isolate growth plate tissue from either proliferative or hypertrophic zone, isolated RNA, and performed microarray analysis and pathway analysis. We found that 49 genes were downregulated ($P < 0.001$, fold-change > 1.3) in PZ of Ezh1/2 mice while 231 genes were upregulated compared to wild type. Similarly in HZ, 97 genes were downregulated ($P < 0.001$, fold-change > 1.3) and 146

genes were upregulated in the Ezh1/2 mice, compared to the wild type mice (Fig 5a, c). To confirm the validity of the microarray analysis, we performed real-time PCR on a subset of genes that showed differential regulation (Fig. 5e). The observation that more genes were upregulated in Ezh1/2 mice (as opposed to downregulated), both in PZ and HZ, is consistent with the role of Ezh1 and 2 in catalyzing H3K27 trimethylation, which serves as a repressive mark of gene expression. Interestingly, the number of genes that were upregulated or downregulated concordantly in both the PZ and the HZ was modest (Fig. 5b). Thus the targets of H3K27 repression appear to depend on the differentiation state of the chondrocytes. The microarrays results also confirmed the upregulation of Cdkn2a/b in PZ and the upregulation of Igfbp3/5 in HZ (supplemental data). Next, we used Ingenuity Pathway Analysis to explore additional potential signaling pathways (Fig. 5d). As expected, this analysis implicated cell-cycle G1/S regulation in the PZ (which includes upregulation of Cdkn2a/b) and implicated IGF signaling in HZ (which includes upregulation of Igfbp3/5). In addition, the analysis implicated Wnt/beta-catenin and glucocorticoid receptor signaling in PZ and thyroid hormone receptor and retinoid (RAR) signaling, all of which are thought to play important roles in the regulation of growth plate chondrocytes.”

Specific Comments

Reviewer comment: [Figure 2] It is unclear from the images in panel g/h that hypertrophic chondrocyte size is altered in KO animals versus controls. Is this measurement statistically significant?

Response: The size of the hypertrophic chondrocyte is altered particularly in the axis parallel to the long axis of the bone, that is, its “height” which is the vertical dimension in Fig. 2g, h. This dimension is particularly important because it is a major determinant of the bone growth rate (Kember NF, Walker KV, *Nature* 1971). This height measurement is quantified and presented in Fig. 2j. The difference between the Ezh1/2 mice and wild-type mice was statistically significant ($P < 0.05$ for all 3 time points studied). To clarify this issue, we have added brackets to Fig 2g and h to indicate cell height. The P-value was included in Fig 2j, bottom panel, and we have amended the text to indicate that we are measuring the height of the hypertrophic cell and explained why this dimension is of particular interest:

“In the combined knockout mice, the number of HZ chondrocytes per column was increased but the height (dimension parallel to the long axis of the bone) of the terminal HZ chondrocyte was diminished (Fig. 2G-H, 2J, $P < 0.05$). The height of the terminal HZ chondrocyte plays an important role in determining the rate of longitudinal bone growth (13), and therefore the observed decrease in height of hypertrophic chondrocytes, which persisted in older mice (Fig. 2G-H and 2J), likely contributed to the impaired bone growth).”

Reviewer comment: [Figure 3] Knockdown experiment of Cdkn2a/b should be accompanied by a Western blot to determine relative protein levels upon knockdown.

Response: As the reviewer suggested, we have confirmed the downregulation of Cdkn2a and b protein level by siRNA using western blot (see Fig. S6).

Reviewer comment: Additionally, it is not clear why Gapdh, a gene presumably not repressed/marked by H3K27me3 was used as a control in this experiment. What exactly does this control for? IgG is a sufficient negative control. A lineage restricted gene would serve as a much better indicator in this experiment as it should maintain repression during chondrogenesis.

Response: We agree that IgG was the most important negative control. A higher ChIP signal using antibody against H3K27me3 compared to IgG indicates presence of H3K27me3. But when we compare the ChIP signal between wild-type cells and Ezh1/2 knockout cells, we think including a negative control that is not known to be enriched with H3K27me3 mark (such as Gapdh) is useful. We found that the promoters of all four of our target genes, Cdkn2a/b and Igfbp3/5, had higher ChIP signal in the wild-type cells compared with knockout cells. Two alternative hypotheses could potentially explain this observation. One possibility is that H3K27me3 is enriched in specific promoters in wild-type cells, but lost in the knockout cells. Alternatively, it could be that H3K27me3 is higher not just in these particular promoters but globally, compared with the knockout cells. The fact that we studied Gapdh and observed low signal in both wildtype and knockout cells argues against the second explanation.

Reviewer #3

Summary

This study reports at a very novel and significant finding demonstrating that the contribution of the Ezh enzymes to chondrogenesis. This is an important paper for several reasons, 1.) It's well known that only this enzyme family produces the repressive histone modification mark H3K27me3, and 2.) Knockdown of either Ezh1 or Ezh2 alone has no effect on skeletal development; however, the double KO (Col2 cartilage-specific cre Ezh1^{-/-}Ezh2 fl/fl) has severely impaired overall growth of the mouse. A continuing decrease in the columnar proliferation zone from 3 to 14 days age is observed and consequently, the same proportional decrease in the pre hypertrophic zone. Mechanisms contributing to decelerated maturation of chondrocytes forming the growth plate zones and thus endochondral bone formation (EBF) were established.

The authors demonstrated in a series of rigorous studies using several approaches a proliferation defect, an increase in H3K27me3 and increased Cdkn2a, which is target of the PRC2 complex that includes EZH2. Confirmation of this mechanism was validated by analyses of the gene in isolated cartilage zones (by LCM) and by ex vivo chondrocyte culture treated with Ezh1/2 inhibitors. The authors then examined deregulated IGF signaling contributing to the diminished HZ, as IGF impaired signaling would account for retarded overall growth. Again rigorous studies identified two highly upregulated IGF inhibitor binding proteins (BP3, BP5) with no change in expression of IGF or IGF1R. Strong evidence is presented including rescue of the phenotype. In vitro and in vivo.

Suggested Improvements:

While there is enthusiasm for this study, and the mice will surely be wanted numerous laboratories, there are still questions that can be readily addressed.

Reviewer suggested improvement 1: Although the mechanism for inhibited proliferation and maturation of the HZ are validate, there are indications from the histological sections that more is going on. Two of the critical regulatory factors distinctly expressed in each zone that regulate the timing of chondrocyte maturation for EBF, namely PTHrP and IHH, are involved in intricate feed -back and forward signaling that control differentiation. Type X collagen and IHH are decreased in cultured chondrocytes (Fig 4g), but perhaps an immunohistochemical staining would better identify the molecular signature of the in vivo HZ cells. The fact that the DKO HZ cells are smaller at day 14 suggests they are the IHH signaling cells. Performing an IHC and also Type X IHC would clarify the block in differentiation.

Response: The reviewer suggests a very interesting hypothesis, that the small cells in the hypertrophic zone of the DKO mice might be prehypertrophic, indicating a block in further differentiation. To test this hypothesis, as suggested by the reviewer, we performed in situ hybridization in wild-type and DKO growth plate to identify the location of *Ihh* and *ColX* expression. We found that the relatively small cells in the hypertrophic zone of DKO mice sequentially expressed *Ihh* then *ColX* in a pattern similar to that seen in wild-type mice, suggesting that, although most of the hypertrophic cells are small in the DKO, they are not *Ihh*-expressing prehypertrophic cells, but rather, fully differentiated *ColX*-expressing cells. The data are now included as Fig 2k-n, and we have added the following to the result section:

“To determine whether the smaller cells in the hypertrophic zone of the *Ezh1/2* mice might represent prehypertrophic chondrocytes, we used in situ hybridization to determine the location of *Ihh* (a marker for prehypertrophic chondrocytes) and *ColX* (a marker for hypertrophic chondrocytes) expression (Fig. 2k-n). We found that these smaller cells in the hypertrophic zone of *Ezh1/2* mice expressed *ColX*, rather than *Ihh*, in a pattern similar to that seen in wild-type mice, suggesting that although most of the cells in the HZ are small in the *Ezh1/2* mice, they are fully differentiated hypertrophic cells.”

Reviewer suggested improvement 2: The source of the lack of clarity can be solved by showing images of the full length bone, for Fig 2 and Fig 4 particularly panel L and O. (these can be in supplement) as a supplement.

Response: We have now included images of whole epiphysis (Fig S4) and metatarsal growth plate (Fig S9) as supplemental figures.

Reviewer suggested improvement 3: Figure 2- the growth plate zones should be labeled on the panels: RZ for resting zone, PZ proliferating zone and more accurate labeling of the hypertrophic zones for, HZ (mature, hypertrophic cells cartilage not calcified) and terminal HZ (calcified cartilage). There is also a Pre -HZ that is not easy to see at day 7, but on day 14 is evident only in the WT mouse and extends a 2-3 mm below the lower dotted line. In striking contrast, that transition is not present in DKO mice, but clearly shows you have Hypertrophic Zone of uncalcified cartilage and a terminal HZ calcified cartilage zone which is Mallory Trichrome stained and weakly in DKO bone.

Response: As suggested by the reviewer, we have now labeled the RZ, PZ, and HZ in the figures. The sections shown in Fig. 2 were all obtained from bones that had been decalcified to facilitate histological sectioning and staining. Therefore, we are unable to distinguish the region of the hypertrophic zone that contains calcified matrix from the region that contains uncalcified matrix. We appreciate the reviewer's observation that the prehypertrophic zone appears less prominent in the 14-day old DKO mice compared to wild-type mice. However, we focused primarily on younger mice because, by 14 days of life, the mice show signs of respiratory distress, which may affect skeletal growth and therefore complicates the data interpretation.

Reviewer suggested improvement 4: Size concerns. Figure 1 shows mouse half the size at 1 wk, but growth chart very close at 1 week.

Response: As the reviewer points out, the difference in body size of the mice in Fig 1b appears greater than the average difference in body weight shown in Fig 1b or the difference in tibial length shown in Fig. 2i. We have therefore bred more mice and photographed a pair of mice that more closely approximates the average difference in size at this age.

Reviewer suggested improvement 4 (continued): Figure 1i lower limbs - are missing? Were these removed for analyses or are they not developed at day 3.

Response: As the reviewer noticed, we removed the lower limbs from these 3-day old mice to perform microdissection and RNA extraction. Since the main purpose of showing the whole mount at 3 days of age was to demonstrate the abnormal curvature of the spine, we are now only showing the spine, rather than the whole animal, to avoid any confusion due to the missing limb.

Reviewer suggested improvement 4 (continued): Figure 2 at 14 day shows an epiphysis with an articular surface and a little bit of joint. There is no secondary center of ossification. In full sections of the limb did the epiphysis change in proportion to bone size. Again having full length sections shown so that the primary spongiosa can be visualized would be informative for understanding the phenotype especially after treatment with IGF

Response: As mentioned above, we have now included images of whole epiphysis as Fig S4.

Reviewer suggested improvement 5: A ChIP was performed with H3K27 me 3. This ChIP could have been analyzed for ColX and IHH - as nothing is known related to their repression by epigenetic regulators.

Response: As suggested by the reviewer, we have analyzed ColX and Ihh promoters in our ChIP samples and the results are now included as Fig S8.

Discussion and References:

With only 4 papers in the literatures related to the author's studies, they could have been indicated. Particularly worthy of comparison in Discussion is the cartilage phenotype resulting from homozygous deletion of K27me3 by the demethylase JMJD3.

Response: We have added the following to the discussion:

“Several prior studies have investigated the role of Ezh2 and H3K27 methylation in growth plate chondrogenesis. Dudakovic et al. studied a mouse with Ezh2 ablated in uncommitted mesenchymal cells (9). Thus their model differed from ours in that loss of Ezh2, occurred earlier in cell differentiation and Ezh1 was left intact. They observed shortened long bones and vertebrae, reduced hypertrophic zone height and overall growth plate height. Zhang et al studied Jmjd3, which encodes an H3K27me3 demethylase and found that ablation of the gene in mice decreases proliferation and delays hypertrophy of chondrocytes, indicating that increased H3K27 methylation, like the decreased methylation in the current study, can severely impair endochondral bone formation (35). Very recently, Mirzamohammadi et al described a mouse with cartilage-specific loss of Eed, which like Ezh2, is a component of the Polycomb repressive complex 2 that catalyzes trimethylation of H3K27 (22). These mice showed effects similar to Ezh1/2 mice with kyphosis, decreased chondrocyte proliferation, accelerated hypertrophic differentiation and cell death with reduced Hif1a expression. Their study focused on the roles of excessive Wnt and TGF- β signaling.”

The last paper from Mirzamohammadi et al was published in June, 2016, while our study was under review.

REVIEWERS' COMMENTS:

Reviewer #1 (Remarks to the Author):

The authors have completed some of the extra studies required and where not, a plausible explanation has been offered. Also, where appropriate, they have softened the conclusions drawn. These additions and edits have improved the manuscript considerably

Reviewer #2 (Remarks to the Author):

Accept

Reviewer #3 (Remarks to the Author):

The authors have made extraordinary efforts in obtaining additional data at multiple levels, particularly performing the global profiling further strengthens their findings. They also have clarified many details that provide better insight into the findings and makes the EZH1/2 study more impactful for developmental biologist examining other tissues. The manuscript is also more reader friendly to those who are not familiar with endochondral bone formation as a model for epigenetic regulation of development.